



SOIL

# Iron and aluminum association with microbially processed organic matter via meso-density aggregate formation across soils: organo-metallic glue hypothesis

**Rota Wagai[1], Masako Kajiura[1], and Maki Asano[2]**

[1]National Agriculture and Food Research Organization, Institute for Agro-Environmental Sciences,
3-1-3 Kan-nondai, Tsukuba, Ibaraki, 305-8604, Japan
[2]Faculty of Life and Environmental Sciences, University of Tsukuba,
1-1-1 Tennodai, Tsukuba, Ibaraki 305-8572, Japan

**Correspondence:** Rota Wagai (rota@affrc.go.jp)

**Abstract.** Global significance of iron (Fe) and aluminum (Al) for the storage of organic matter (OM) in soils and surface sediments is increasingly recognized. Yet specific metal phases involved or the mechanism behind metal–OM correlations frequently shown across soils remain unclear. We identified the allocation of major metal phases and OM to density fractions using 23 soil samples from five climate zones and five soil orders (Andisols, Spodosols, Inceptisols, Mollisols, Ultisols) from Asia and North America, including several subsurface horizons and both natural and managed soils. Each soil was separated into four to seven density fractions using sodium polytungstate with mechanical shaking, followed by the sequential extraction of each fraction with pyrophosphate (PP), acid oxalate (OX), and finally dithionite–citrate (DC) to estimate pedogenic metal phases of different solubility and crystallinity. The concentrations of Fe and Al (per fraction) extracted by each of the three reagents were generally higher in meso-density fractions ($1.8$–$2.4\,\mathrm{g\,cm^{-3}}$) than in the lower- or higher-density fractions, showing a unique unimodal pattern along the particle density gradient for each soil. Across the studied soils, the maximum metal concentrations were always at the meso-density range within which PP-extractable metals peaked at $0.3$–$0.4\,\mathrm{g\,cm^{-3}}$ lower-density range relative to OX- and DC-extractable metals. Meso-density fractions, consisting largely of aggregated clusters based on SEM observation, accounted for on average 56 %–70 % of total extractable metals and OM present in these soils. The OM in meso-density fractions showed a 2–23 unit lower C : N ratio than the lowest-density fraction of the respective soil and thus appeared microbially processed relative to the original plant material. The amounts of PP- and OX-extractable metals correlated positively with co-dissolved C across the soils and, to some extent, across the density fractions within each soil. These results led to a hypothesis which involves two distinct levels of organo-metal interaction: (1) the formation of OM-rich, mixed metal phases with fixed OM : metal stoichiometry followed by (2) the development of meso-density microaggregates via "gluing" action of these organo-metallic phases by entraining other organic and mineral particles such as phyllosilicate clays. Given that OM is mainly located in meso-density fractions, a soil's capacity to protect OM may be controlled by the balance of three processes: (i) microbial processing of plant-derived OM, (ii) dissolution of metals, and (iii) the synthesis of organo-metallic phases and their association with clays to form meso-density microaggregates. The current hypothesis may help to fill the gap between well-studied molecular-scale interaction (e.g., OM adsorption on mineral surface, coprecipitation) and larger-scale processes such as aggregation, C accrual, and pedogenesis.

Published by Copernicus Publications on behalf of the European Geosciences Union.

# 1  Introduction

Organic matter (OM) stored in soil plays a fundamental role in ecosystem functioning through the storage of carbon (C) and nutrients, improvement of aeration and water-holding capacity, and thus plant productivity and biogeochemical cycling. Changes in soil OM have a significant impact on future climate as soil represents the largest terrestrial C pool. The storage capacity and stability of soil OM are particularly important questions for our efforts to limit global warming (Smith, 2016). Soil OM stability is strongly controlled by its association with soil minerals via chemical interaction and physical aggregation (Lehmann and Kleber, 2015; Sollins et al., 1996). The mineral parameters often used to estimate soil's protective capacity are clay content ($< 2\,\mu m$) or clay plus silt content ($< 20\,\mu m$) of soils as they often correlate with soil OM contents, and these small-sized minerals tend to have a high surface area to adsorb OM (Six et al., 2002, and the references therein). Commonly used mathematical models to predict soil C changes use these parameters to slow down OM turnover and to increase its storage (Coleman and Jenkinson, 1996; Parton et al., 1987; Wieder et al., 2015).

On the other hand, the global significance of iron (Fe) and aluminum (Al) phases for OM storage in soil and surface sediments has been increasingly recognized. Using 5500 pedons around the world, Rasmussen et al. (2018) showed stronger control of organic C storage in non-arid soils by oxalate-extractable metal content than by clay content. Important linkage among climate (especially water balance), dissolved organic C production, and its stabilization by these metal phases has been shown on a continental scale (Kramer and Chadwick, 2018). Surface marine sediments also store significant amounts of Fe-bound C (Lalonde et al., 2012). Iron and aluminum, the third and fourth most abundant elements on the earth crust, are in fact highly reactive with OM once released via chemical weathering. These pedogenic Fe and Al can be present in monomeric form by chelating with organic ligands or in polymeric form as polynuclear complexes and as secondary minerals. The latter includes Fe and Al oxides, hydroxides, and oxyhydroxides (collectively called metal oxides, hereafter) as well as short-range-order aluminosilicates (allophane, imogolite, and proto-imogolite) that have high sorptive capacity for OM due to their small size (down to several nanometers) and high surface reactivity via surface hydroxyl groups (Fuji et al., 2019; Kaiser and Guggenberger, 2003; Kleber et al., 2015). In addition, soluble complexes of Fe and Al with organic ligands can be precipitated especially in acidic, OM-rich environments such as volcanic and podzolic soils (Lundström et al., 2000; Percival et al., 2000; Takahashi and Dahlgren, 2016).

Incorporating such metal control into soil C models is still a challenge because the mechanisms by which pedogenic metals control OM storage and stabilization remain elusive. This is partly because current understanding relies largely on OM–metal correlations where the metal concentration often co-varies with other soil properties. The reactive metal contents often positively correlate with other mineralogical parameters that are considered to contribute to OM storage such as clay content and soil-specific surface area (e.g., Kaiser and Guggenberger, 2003; Mayer and Xing, 2001). In long-term pedogenesis (240–4100 kyr) under a temperate or tropical moist climate regime, radiocarbon-based soil C age was positively correlated with extractable metal contents in two chronosequence studies (Masiello et al., 2004; Torn et al., 1997) but not in another study under wetter climate where only soil-specific surface area and halloysite content showed significant correlation (Lawrence et al., 2015). Short-range-order minerals and Fe oxides can also promote aggregation (Churchman and Tate, 1986; Oades and Waters, 1991; Shang and Tiessen, 1998), which indirectly enhances OM stability (Balesdent et al., 2000; Totsche et al., 2017) without necessarily showing proportionality to metal concentrations.

To untangle co-varying factors, Wagai and Mayer (2007) assessed Fe oxide contribution to C storage by quantifying the C released during the reductive Fe oxide dissolution with dithionite for soils covering eight soil orders. After correcting for the OC release due to salt and extractive pH effects, the study conservatively estimated that 2 %–25 % (up to 60 % for a highly weathered, Fe-rich soil) of total soil OM was Fe-bound and the C : Fe ratio of the extracts suggested greater contribution of precipitated organo-metal complexes than simple adsorptive association in lower-pH soils. Subsequent dithionite-based studies confirmed that less than half (typically less than a quarter) of total OM was associated with Fe in a range of soils and sediments (Coward et al., 2018, 2017; Lalonde et al., 2012; Zhao et al., 2016). Wagai et al. (2013) further examined the potential contribution of other metal phases such as short-range-order minerals, organo-metal complexes, and their coprecipitates using other extractants and showed that, even as liberal estimates, 5 %–60 % of total OM in a range of acidic to near-neutral soils (higher percentages for volcanic soils and spodic horizons) could be explained by direct association with Fe and Al phases. The limited extractability of OM with these metal phases implied the potentially critical role of physical protection via ternary associations of OM, metals, and clay (Wagai and Mayer, 2007). More recent studies showed that a portion of soil Fe phases in soil can be protected from reductive dissolution (dithionite extraction) due presumably to physical protection within microaggregates or coprecipitation with short-range-order aluminosilicates (Coward et al., 2018; Suda and Makino, 2016; Filimonova et al., 2016), suggesting both the limitation of a single-extraction approach and the importance of aggregation–precipitation reactions. On the other hand, experimental studies revealed specific factors and underlying mechanisms behind OM–metal interaction via adsorption, complexation, and coprecipitation using pure metal phases under well-defined laboratory conditions (Chen et al., 2014; Mikutta et al., 2011; Nierop et al., 2002; Schneider et al., 2010; Tamrat et al., 2019; Kaiser, 2003).

https://doi.org/10.5194/soil-6-1-2020

However, a large knowledge gap still remains between the laboratory studies and field-based studies (see a review by Kleber et al., 2015).

Critical to filling this gap is to identify factors controlling the distribution and localization of different pedogenic metal phases in soil systems because different modes of organo-metal association likely take place at specific local environments. At larger spatial scales, mobilization and accumulation of pedogenic metal phases and their interaction with OM over time and along soil profile depth have been well-recognized (e.g., Kramer et al., 2012; Lawrence et al., 2015). At micro- to nano-meter scales, on the other hand, advanced imaging techniques revealed co-localization of OM and pedogenic Fe and Al in natural soils (Asano et al., 2018; Garcia Arredondo et al., 2019; Inagaki et al., 2020; Wan et al., 2007), although up-scaling of such information remains a major challenge. At horizon or bulk soil scales, physical fractionation studies have indicated the presence – but not the spatial arrangement – of multiple OM pools of varying turnover rate and degree of mineral associations (Christensen, 2001; von Lützow et al., 2007).

Density provides a useful fractionation approach to assess the localization of pedogenic Fe and Al phases because many biogeochemical characteristics of soil particles are closely related to their density (Christensen, 2001; Sollins et al., 2009; Turchenek and Oades, 1979; von Lützow et al., 2007). The density of soil particles is primarily controlled by the relative amounts of the two major components, OM and mineral. The average density of OM present in soil is typically around $1.4 \, \mathrm{g \, cm^{-3}}$ but can range between 1.1 and $1.9 \, \mathrm{g \, cm^{-3}}$ across soils and sediments (Mayer et al., 2004). The minerals typically found in soils such as aluminosilicate clays have the density of $> 2.4 \, \mathrm{g \, cm^{-3}}$ including short-range-order allophane and imogolite (2.75 and $2.70 \, \mathrm{g \, cm^{-3}}$, Wada, 2018) with minor exceptions such as phytoliths ($2.1–2.15 \, \mathrm{g \, cm^{-3}}$, Drees et al., 1989). Among metal oxides, Fe oxides have much higher densities ($3.8–5.3 \, \mathrm{g \, cm^{-3}}$, Cornell and Schwertmann, 2003) compared with Al hydroxides such as gibbsite ($2.4 \, \mathrm{g \, cm^{-3}}$, Anthony et al., 1997). Pedogenic metals can remain as distinct phases or they can associate with OM or other minerals. From the perspective of OM, both pedogenic metals and other minerals will be found in a high-density fraction ($> 2.4 \, \mathrm{g \, cm^{-3}}$) unless they associate with OM. While OM binding lowers the density of Fe and Al phases in the laboratory (Kaiser and Guggenberger, 2007), its extent under various pedogenic environments is virtually unstudied.

We hypothesize that most pedogenic metals are found in $< 2.4 \, \mathrm{g \, cm^{-3}}$ density fractions regardless of soil type due to their high reactivity with OM. Because of the higher density and redox sensitivity, pedogenic Fe may concentrate in different density phases from pedogenic Al for any given soil, assuming that organo-Fe and organo-Al associations take place independently. Alternatively, if the metal dissolution and subsequent interaction with OM are regu-

lated by the same environmental factors, their distribution along the particle density gradient would be similar. Furthermore, if submicron-sized, OM-rich metal phases act as binding agents as suggested for a volcanic soil based on STXM/NEXAFS and electron microscopy (Asano and Wagai, 2014; Asano et al., 2018), they may preferentially bind with other fine-sized minerals (e.g., clays) to form the ternary associations previously postulated (Wagai and Mayer, 2007). These ideas were tested by fractionating soil particles into four to seven density classes after a mild level of dispersion (mechanical shaking in solution) and by quantifying the amounts of Fe and Al phases by selective dissolution techniques that target different metal phases (pyrophosphate, acid oxalate, and dithionite–citrate extractions in sequence) using 23 soil samples from 11 sites spanning 5 climate zones and 5 soil orders (Andisols, Spodosols, Inceptisols, Mollisols, Ultisols) and including several subsurface horizons and both natural and managed (upland and paddy) soils.

Terminology: weathering products of Fe- and Al-bearing minerals during pedogenesis are collectively called "pedogenic Fe and Al phases" in this study. We use the term "complex" to refer to aqueous organo-metal complexes and their precipitates and avoid its use to describe broader associations such as organo-clay and organo-mineral complex. Various types of OM–metal association are discussed by grouping into three general mechanisms (adsorption, complexation, and aggregation) while recognizing that organo-metal complexes in soil are mainly coprecipitated with other metal phases and particles. The term "particle" is used in a broad sense to include aggregates as well as single organic or mineral particles.

## 2 Methods

### 2.1 Soil sample source

The soil samples selected for this study reflect our primary interest in the soils and soil horizons that hold high OM via its interaction with reactive mineral phases. A soil sample set consisted of four groups: allophanic Andisol (silandic), non-allophanic (aluandic) Andisol, spodic, and crystalline mineralogy ($n = 23$, Table 1) groups, including both natural and cultivated soils from six climate zones with a wide range of mean annual temperature (5–24 °C) and precipitation (221–2392 $\mathrm{mm \, yr^{-1}}$).

Allophanic Andisol samples were collected in the Kanto plain and Kofu basin, Japan. Parent material is mainly rhyolitic and basaltic volcanic ash deposits. Dominant clay-sized minerals are short-range-order (SRO) minerals – more allophane-/imogolite-type minerals than hydrous iron oxides such as ferrihydrite. Minor amounts of gibbsite, kaolinite, chlorite, hydroxyl-interlayered vermiculite, mica, quartz, and feldspar are often found in these soils. Five of the samples were from a long-term field experiment for OM management (A-1 to A-5) and the data from these samples have been re-

**Table 1.** Sample source information and basic soil characteristics.

| Sample ID | Site | Climate zone[a,b] | MAT °C | MAP mm yr$^{-1}$ | Soil order[c] | Land use[a] | Vegetation/management | Horizon/depth (cm) | TOC mg g$^{-1}$ | TN mg g$^{-1}$ | Soil pH (H$_2$O) | Clay (%) |
|---|---|---|---|---|---|---|---|---|---|---|---|---|
| **Andisols, allophanic** | | | | | | | | | | | | |
| A-1[d]/NIA-NTa | Tsukuba, Japan | WTM | 13.7 | 1300 | Andisol | Cropland | No-till with leaf compost | Ap/0–5 | 149 | 10 | 6.2 | 45–55 |
| A-2[d]/NIA-NTb | | | | | | | | Ap/5–20 | 80.4 | 5.9 | 6.2 | 45–55 |
| A-3[d]/NIA-Till1 | | | | | | | Till + NPK | Ap/0–20 | 51.4 | 4.1 | 6.1 | 45–55 |
| A-4[d]/NIA-Till2 | | | | | | | Till (stump removed) | Ap/0–20 | 42.3 | 3.6 | 6.7 | 45–55 |
| A-5/NIA-Bare | | | | | | | Bare fallow | Ap/0–20 | 36.9 | 3.0 | 6.5 | 45–55 |
| A-6/TKY-NPK | Tokyo, Japan | WTM | 14.7 | 1515 | Andisol | Cropland | Till + NPK | Ap/0–14 | 46.5 | 3.9 | 4.1 | 27 |
| A-7/TKY-Comp | | | | | | | Till + NPK + compost | Ap/0–14 | 60.3 | 5.4 | 5.0 | 28 |
| A-8/YMS-NPK | Yamanashi, Japan | WTM | 12.1 | 1137 | Andisol | Cropland | Till + NPK | Ap/0–14 | 29.1 | 2.8 | 6.6 | 28 |
| A-9/YMS-Straw | | | | | | | Till + NPK + rice straw | Ap/0–14 | 35.5 | 3.2 | 6.7 | 28 |
| A-10/MTT-A | Tsukuba, Japan | WTM | 13.1 | 1438 | Andisol | Forest | Secondary forest (mixed) | A/0–15 | 97.7 | 7.6 | 4.6 | 35–45 |
| **Andisols, non-allophanic** | | | | | | | | | | | | |
| N-1/KWT-A1 | Kawatabi, Japan | WTM | 10.1 | 1460 | Andisol | Grassland | Semi-managed | A1/0–25 | 105 | 6.7 | 4.6 | 23 |
| N-2/KWT-A2 | | | | | | | | A2/25–52 | 102 | 5.6 | 4.6 | 18 |
| N-3/KWT-2A3 | | | | | | | | 2A3/52–100 | 98.3 | 4.9 | 4.9 | 25 |
| N-4/KWT-3Bw | | | | | | | | 3Bw/120–140 | 12.1 | 0.9 | 5.2 | 11 |
| **Spodosols** | | | | | | | | | | | | |
| S-1/MBB-CN | Maine, USA | CTM | ~5 | 1365 | Spodosol | Forest | Conifer (red spruce) | Bhs/5–15 | 146 | 5.8 | 3.7 | 18–35 |
| S-2/MBB-DD | | | | | | | Mixed hardwood | Bhs/7–22 | 58.9 | 2.4 | 4.6 | 18–35 |
| **Crystalline mineralogy soils** | | | | | | | | | | | | |
| C-1/NGK-Manure | Aichi, Japan | WTM | 14.9 | 1447 | Inceptisol | Cropland | Rice paddy + manure | Ap/0–15 | 24.1 | 2.1 | 5.9 | 15–25 |
| C-2/WKY-NPK | Wakayama, Japan | WTM | 15.3 | 1286 | Ultisol | Cropland | Till + NPK | Ap/0–16 | 9.8 | 1.0 | 5.8 | 15–25 |
| C-3/WKY-Straw | | | | | | Cropland | Till + NPK + manure | Ap/0–16 | 28.8 | 2.5 | 6.4 | 15–25 |
| C-4/MNG-A | NE Mongolia | CTD | ND | 221 | Mollisol | Grassland | Forest steppe | A1/0–10 | 24.2 | 2.5 | 6.4 | 15–25 |
| C-5/KIN-07S | Saba, Malaysia | TW | 24.0 | 2392 | Ultisol | Forest | Tropical (hill dipterocarp) | A/0–10 | 20.9 | 1.9 | 3.9 | 33–38 |
| C-6/KIN-17S | Saba, Malaysia | TM | 19.0 | 2380 | Inceptisol | Forest | Tropical (lower montane) | A/0–10 | 42.0 | 2.5 | 4.1 | 10–13 |
| C-7/KIN-27S | Saba, Malaysia | WTM | 13.0 | 2256 | Inceptisol | Forest | Tropical (upper montane) | A/0–10 | 125 | 8.4 | 3.9 | 6–15 |

[a] Based on 2006 IPCC Guidelines for National Greenhouse Gas Inventories. Vol. 4, Chapter 3: Consistent Representation of Lands. TS2 [b] For climate zone, WTM: warm temperate moist, CTM: cool temperate moist, CTD: cool temperate dry; TW: tropical wet, and TM: tropical montane. [c] Soil taxonomy, USDA, soil survey staff. [d] TOC, TN data from Wagai et al. (2018).

ported previously (Wagai et al., 2018). The sample group also includes two pairs of soils (A-6 vs. A-7, and A-8 vs. A-9) from other long-term field experiments of OM amendment and one soil from a relatively undisturbed secondary forest site.

Non-allophanic Andisol samples consist of four horizons from a well-characterized pedon (Pacllic Mdanudand) from the Field Science Center of Tohoku University, Miyagi, Japan, located on a gentle slope of a fan in mountain valleys at 190 m elevation maintained as grassland (Sasa nipponica). Parent material is dacitic volcanic tephra with alluvium including smectitic sedimentary rock. A2 horizon contained a key tephra (Hijiori pumice: 10 kyr BP).

Spodic group consists from the spodic horizon from two pedons (coarse-loamy, isotic, frigid, Typic Haplorthods, developed from ca. 1 m thick glacial till) under coniferous and deciduous forest types in Bear Brooke Watershed, Maine, USA. Further details on the sites and soil OM characteristics can be found in Ohno et al. (2017). Major soil minerals are quartz with moderate amounts of plagioclase, K-feldspar, and hornblende (Swoboda-Colberg and Drever, 1993).

Crystalline mineralogy group consists of a range of soils relatively low in extractable Fe and Al phases or high in more stable clay-sized minerals such as kaolinite. The paddy soil (C-1) has been under seasonal flooding (May to August/September) for more than several decades. The other two soils (C-2, C-3, classified as Aquic Hapludult) are from long-term experiments of manure application. The most arid soil in our sample set is Mongolian forest steppe soil (C-4), Calcic Kastanonzems in the WRB classification system. Our A horizon sample, however, contains no carbonate (Asano et al., 2007 TS3). The other three soils are from an elevation gradient under tropical rainforest with the dominant clay mineralogy of kaolinite, gibbsite, hydroxyl-interlayered vermiculite, and quartz (C-5) and illite, kaolinite, and quartz (C-6, C-7). More details on the sites and soils are shown in Wagai et al. (2008) and Tashiro et al. (2018).

All samples were air-dried and 2 mm sieved prior to density fractionation and chemical analyses. Air-drying did not significantly change OM and metal distribution across density fractions, showing no irreversible aggregation by air-drying for an allophanic Andisol (Wagai et al., 2015) and presumably for the other soils. Carbon refers to organic C in this study as no carbonate was found in these soils.

## 2.2 Physical fractionation by density

We sorted soil particles based on particle density using sodium polytungstate (SPT-0 grade, Sometsu, Germany) to make the liquids of various densities. We employed mechanical shaking to disrupt less-stable aggregates and sequentially separated density fractions following previous studies (Crow et al., 2014; Sollins et al., 2009; Wagai et al., 2018). Most soil samples were separated into six to seven fractions ($n = 18$), while the other five samples (A-6, A-7, A-8, A-9, C-1) examined at a later stage were fractionated into only four fractions (Table 1, also see Table A1 in Appendix A) because we learned that the main allocation pattern can be captured by four density fractions. The sieved soil samples were mixed with $1.6\,\mathrm{g\,cm^{-3}}$ SPT solution (soil : solution ratio $= 10\,\mathrm{g} : 40\,\mathrm{mL}$), mechanically shaken for 30 min at 120 rpm, and centrifuged (20 min, 2330 g). The floating material ($< 1.6\,\mathrm{g\,cm^{-3}}$, the lowest-density fraction, F1) was collected on a $0.22\,\mathrm{\mu m}$ membrane filter using a vacuum filtration system. These steps (shaking to centrifugation) were repeated three times to maximize the recovery of this fraction. The materials caught on the filter were washed with deionized water until the salt concentration of the final 50 mL of water reached $< 50\,\mathrm{\mu S\,cm^{-1}}$ and then transferred to a beaker for oven-drying at 80 °C. After the isolation of F1, the remaining material in the centrifuge tube was re-suspended in $1.8\,\mathrm{g\,cm^{-3}}$ SPT solution, shaken again, and centrifuged. The floating materials ($1.6$–$1.8\,\mathrm{g\,cm^{-3}}$, F2) were transferred to 250 mL bottles, mixed with deionized water, and centrifuged (17 000 g, 30–60 min), and then the supernatant was discarded. This process was repeated four to five times until the supernatant salt concentration reached $< 50\,\mathrm{\mu S\,cm^{-1}}$ and the rinsed materials were freeze-dried. Following the same procedure, we then sequentially isolated higher-density fractions (e.g., F3: 1.8–2.0, F4: 2.0–2.25, F5: 2.25–2.5, and F6: $> 2.5$) using correspondingly higher-density SPT solutions. To fully recover each density fraction, we repeated the steps (from shaking to the recovery of floating materials) at least three times.

The selection of cutoff density for higher-density fractions varied among the soils (Table A1). This was partly due to soil mineralogical difference (e.g., soils expected to have higher Fe contents had the highest-density cutoff of 2.75 instead of $2.6\,\mathrm{g\,cm^{-3}}$). The lowest-density fractions ($< 1.6\,\mathrm{g\,cm^{-3}}$) were oven-dried at 80 °C instead of freeze-drying for logistical reasons. Due to the concentration of the extractable metals in this fraction, we assumed little effect of the difference in the drying method on our result interpretation. We also assume little impact of sodium polytungstate on the extractability of Fe and Al phases or the nature of soil microaggregates as the SPT solution after the density fractionation typically had a pH value similar to bulk soil pH.

## 2.3 Extraction of metal phases

Bulk and density-fractionated samples were sequentially extracted by sodium pyrophosphate (PP) followed by acid oxalate in the dark (OX) and then by dithionite–citrate (DC) following Wagai et al. (2018). First, initial PP extraction was conducted at the soil : solution ratio of 100 mg : 10 mL with 0.1 M sodium pyrophosphate (pH $= 10$) and then shaken at 120 rpm for 16 h. After high-speed centrifugation (29 000 g, 45 min), an aliquot of the extract was immediately taken for dissolved C, N, and metal analyses. Second, the residue after discarding the remaining supernatant was re-suspended

and extracted with 10 mL of 0.2 M acidified sodium oxalate solution (pH = 3.0), shaken at 120 rpm for 4 h in the dark. The conventional acid oxalate method (Loeppert and Inskeep, 1996) was modified by replacing ammonium oxalate with sodium oxalate to allow the direct quantification of co-dissolved N while achieving the same extraction efficiency of Fe, Al, and Si (Wagai et al., 2013). After the high-speed centrifugation, an aliquot of the extract was immediately diluted for metal and N analyses to avoid precipitation. Third, 0.1 g of sodium dithionite was added to the remaining residue and mixed with 10 mL of 22 % (by wt) sodium citrate. The mixture was shaken for 16 h and centrifuged under the same condition as above. All extractions were done at room temperature (20–22 °C). We did not filter the supernatants after the high-speed centrifugation of PP extracts as our pilot test showed no systematic decrease in dissolved organic C and metals by vacuum filtration using a 0.025 μm pore-sized membrane (Millipore, VSWP, Bedford, MA, USA). Similarly, no filtration was done for OX and DC extracts after the high-speed centrifugation.

## 2.4 Chemical analyses

The concentrations of Fe, Al, Si, and Mn in the extracts of PP, OX, and DC were analyzed by inductively coupled plasma-optical emission spectroscopy (Vista-Pro, Agilent, CA, USA). The metal analyses were done for all fractions. The only exception is the lowest-density fraction from three soil samples (A-5, N-4, C-2) where low mass recovery prevented the extractions. Analytical errors associated with our density fractionation were sufficiently low for C, N, and the extractable metals (< 12 %, Wagai et al., 2015) to allow testing of our hypothesis. When assessing the role of extractable metal as a whole, we summed weight-based concentrations of Al and Fe as "Al + 0.5 Fe" to approximately normalize the atomic mass difference between Al and Fe for graphical and statistical purposes. This allows us to compare the metal concentration with C on a weight basis. We also reported some values including the stoichiometric relationships among the target elements (e.g., Al : Si ratio) on a molar basis to allow comparison with other literature values.

Co-dissolved organic C and N by PP extraction and the N by OX extraction were quantified by a TOC analyzer (Shimadzu TOC-V/TNM1, Kyoto, Japan). Dissolved organic C (DOC) was measured as non-purgeable organic C after acidification and C-free air purging. Total dissolved N was measured by a chemiluminescence accessory. This method, including the caveats on this technique, was discussed elsewhere (Wagai et al., 2013). Because quantifying the soil C in an oxalate extract is not possible, we estimated the C associated with OX-extractable mineral phases ($DOC_{OX}$) by multiplying total dissolved N concentration in OX extract by the C : N for each density fraction. This estimation assumes that the OM dissolved by the oxalate extraction has the same C : N ratio as that in bulk fraction. This assumption

cannot be fully justified but would be a reasonable approximation for the purpose of assessing the trends because a plot of the C : N ratio of PP-extractable phase against that of bulk soil C : N (regression through the origin) showed a significant positive correlation ($r^2 = 0.89$, $p < 0.0001$) with the slope close to 1 (1.25 with 95 % confidence interval of 1.15–1.32). Total organic C and N concentrations in the isolated fractions and bulk samples were analyzed by an elemental analyzer (Flash2000 Thermo Fisher Scientific Inc., USA). All the elemental concentrations from the three extractions conducted were shown in Table A1.

## 2.5 Extractable elements expressed in two ways

The extractable elements were examined for their concentrations within each density fraction and across the fractions. First, we assessed the *concentration* of the elements for each extractable phase per density fraction mass (e.g., mg Fe g$^{-1}$ fraction). Second, we also assessed the *distribution* of the elements per bulk soil mass (e.g., mg Fe g$^{-1}$ bulk soil) by multiplying the elemental concentration per fraction (mg Fe g$^{-1}$ fraction) by the mass proportion for the respective fraction in bulk soil (g fraction g$^{-1}$ bulk soil).

## 2.6 Peak density determination

For each soil sample, we determined the particle density at which the metal concentration from each extraction was highest (termed "peak density") using two approaches. First, we simply selected the density fraction where the metal concentration was the highest among the fractions for a given soil and used its midpoint (e.g., 2.1 g cm$^{-3}$ for 2.0–2.2 g cm$^{-3}$ fraction) as the peak density. The actual peak density is not necessarily its mean, especially for the samples separated into a smaller number of fractions. Thus, as a second approach, we also estimated the peak density by fitting a normal distribution curve to the metal concentrations per fraction against particle density for each soil (Fig. A1 left panels). The majority of soil samples fitted well, though we had to remove the data points from the lowest- and highest-density fractions in some cases (Fig. A1). For instance, OX- and DC-extractable metal concentrations in the highest-density fractions were quite high for non-allophanic Andisol samples due presumably to magnetite. Thus, those data points were not used for the fitting. Similarly, we eliminated the lowest one or two density fractions of PP- and OX-extractable metal concentration data from six samples from the crystalline mineralogy group. Eliminating these data points (Fig. A1) was done only to enable fitting of a normal distribution (on average $r^2 = 0.90$–0.98, Fig. A2a). The mean of the normal distribution for each extraction/soil was used as the second estimate of peak density. We also determined the mass-weighted particle density at which the metal was most concentrated. After calculating the metal distribution along the density gradient (i.e., metal concentra-

tion × mass fraction), the two approaches above were applied to determine the peak density for each soil sample and each extractable metal phase. A normal distribution fitted well for most samples (Figs. A1 right panels, A2b).

## 2.7 Scanning electron microscopy

Isolated density fractions from selected soil samples were observed by SEM (SU1510, Hitachi high-technologies, Tokyo, Japan). Subsets of freeze-dried density fractions were re-dispersed in ultrapure water by weak sonication ($< 10\,\mathrm{J\,mL^{-1}}$), deposited on carbon tape, and were Pt-Pd coated prior to the observation.

## 2.8 Statistics

Linear regression analyses between extractable metals and DOC were done using JMP software (version 8.0.1, SAS Institute, Cary, NC, USA). Density-dependent change in the proportion of total C in each fraction explained by the association with PP- and OX-extractable phases ($[\mathrm{DOC_{PP}} + \mathrm{DOC_{OX}}] : \mathrm{TC}$) TS4 was modeled for each soil group using polynomial functions using JMP software.

## 3 Results

### 3.1 Recovery after sequential density fractionation

Our density fractionation procedure showed a largely reasonable recovery of mass, C, and N for the 23 soil samples studied (Table 2). While the recovery of the metals assessed by PP-, OX-, and DC sequential extractions was generally good, the variation among the soils was larger (Table 2), as some soil samples and metal phases showed poor recoveries. Possible sources of errors are (i) the small sample mass used for the extractions, (ii) additive errors from the sequential extractions, and (iii) the small sizes of targeted pools. For example, the recovery worsened for DC-extractable Al and Si pools following PP and OX extractions. Incomplete removal of colloidal Fe and Al phases likely explains higher metal recovery from Andisol samples ($118 \pm 36\,\%$) compared to non-Andisol samples ($84 \pm 12\,\%$) in PP extraction. The factors affecting the poor recovery with the sequential extraction approach were discussed in more detail elsewhere (Wagai et al., 2018). We considered the obtained recoveries (Table 2) to be tolerable for assessing the general patterns of the metals and OM across the density fractions.

### 3.2 Concentration of C, N, and extractable metals along the particle density gradient

The C concentration was highest in the lowest-density fraction ($238$–$443\,\mathrm{mg\,C\,g^{-1}}$ fraction), declined with increasing density up to ca. $2.5\,\mathrm{g\,cm^{-3}}$, and remained low ($1.5$–$7.1\,\mathrm{mg\,C\,g^{-1}}$ fraction) at higher densities for each of the studied soils (Fig. A3a). Similarly, C : N ratio showed a progressive decline with increasing density in each soil (Fig. A3b). While the majority of the soil samples showed very low C concentration and C : N ratio in the fractions higher than ca. $2.5\,\mathrm{g\,cm^{-3}}$, a few samples showed a slight increase towards the highest-density fraction.

The extractable metal concentration ($\mathrm{Al} + 0.5\,\mathrm{Fe}$) generally showed a unimodal pattern along the particle density gradient with some distinct patterns among the four soil groups (Fig. 1; also see Fig. A1 left panels for individual soils). The allophanic Andisol group showed higher OX- and DC-extractable metals ($\mathrm{metal_{OX}}$, $\mathrm{metal_{DC}}$), especially in a meso-density (intermediate) range ($1.8$–$2.4\,\mathrm{g\,cm^{-3}}$) up to $78.3\,\mathrm{mg\,g^{-1}}$ fraction ($1.46\,\mathrm{mmol\,g^{-1}}$ fraction) for $\mathrm{metal_{OX}}$ and $27.2\,\mathrm{mg\,g^{-1}}$ fraction ($0.38\,\mathrm{mmol\,g^{-1}}$ fraction) for $\mathrm{metal_{DC}}$, whereas non-allophanic Andisol and Spodosol groups were characterized by high concentrations of PP-extractable metals ($\mathrm{metal_{PP}}$) up to $53.2\,\mathrm{mg\,g^{-1}}$ fraction ($1.97\,\mathrm{mmol\,g^{-1}}$ fraction. As expected, the crystalline mineralogy group showed the lowest levels of $\mathrm{metal_{PP}}$ and $\mathrm{metal_{OX}}$ with moderate amounts of $\mathrm{metal_{DC}}$ from highly weathered soil samples. Two exceptions to the general unimodal pattern were present. For $\mathrm{metal_{PP}}$, three soil samples showed their peak metal concentrations at the lowest density. For $\mathrm{metal_{OX}}$, three samples of non-allophanic Andisols showed the highest metal (especially Fe) concentrations at the highest density due presumably to the presence of Fe-bearing primary minerals such as magnetite. Even in the soils relatively low in specific metal phases (e.g., the crystalline mineralogy group low in $\mathrm{metal_{PP}}$ and $\mathrm{metal_{OX}}$, and the non-allophanic Andisol and Spodosol groups low in $\mathrm{metal_{OX}}$ and $\mathrm{metal_{DC}}$), the highest metal concentrations were found at a meso-density range (Fig. A1 left panels).

Similar patterns were shown when assessing the distribution of the metals along the density gradient by accounting for mass distribution (Fig. A1 right panels). The unimodal pattern remains for most samples, indicating that the major portions of respective metal phases accumulated in a meso-density range. However, the dominant metal phase clearly differed among the soil groups. In the allophanic Andisol group, $63 \pm 9\,\%$ of the total extractable metal was present in OX-extractable phase. On the other hand, the non-allophanic Andisol and Spodosol groups showed that $59 \pm 25\,\%$ and $75 \pm 14\,\%$ of the total extractable metals, respectively, were accounted for by PP-extractable phase. In the crystalline mineralogy group, DC-extractable phase accounted for higher portions ($41 \pm 14\,\%$) of the total extractable metal.

The comparison of extractable Fe and Al among the three extractions generally showed the increasing dominance of low-crystallinity phases (PP- and OX-extractable phases) in lower-density fractions (Fig. A4). The proportion of total extractable Fe present as $\mathrm{Fe_{PP}}$ and $\mathrm{Fe_{OX}}$ showed a clear decline for all four soil groups, with a note that the highest-density fractions in Andisol samples were due presumably to magnetite. A similar declining trend was found for the extractable Al phases. Although minor in quantity, the extractable Mn

**Table 2.** Recoveries of mass, C, N, and the metals dissolved by initial pyrophosphate (PP) and subsequent acid oxalate (OX) and dithionite–citrate (DC) extractions from the density fractions, expressed as percentage relative to the whole soil. Values show means (standard deviations) of all soils and separately for Andisols (allophanic and non-allophanic Andisols) and non-Andisols (Spodosols and crystalline mineral soils).

| Soil group | Mass | C | N | PP extr. | OX extr. | DC extr. | Sum of PP, OX, and DC extractions | | |
|---|---|---|---|---|---|---|---|---|---|
| | | | | Al + 0.5 Fe | Al + 0.5 Fe | Al + 0.5 Fe | Al | Fe | Si |
| All soils ($n = 23$) | 99.8 (2.9) | 97.5 (9.5) | 95.6 (10.6) | 104.6 (33.3) | 98.8 (20.5) | 86.4 (16.9) | 88.8 (16.0) | 94.4 (11.2) | 93.6 (20.4) |
| Volcanic ($n = 14$) | 101.3 (1.9) | 102.2 (6.7) | 101.3 (7.4) | 117.9 (36.0) | 104.9 (22.7) | 86.3 (18.4) | 95.4 (13.4) | 99.3 ( 7.5) | 101.9 (21.5) |
| Non-volcanic ($n = 9$) | 96.8 (2.1) | 90.3 (8.9) | 86.8 (8.9) | 83.9 (12.0) | 89.4 (12.5) | 86.7 (15.5) | 78.7 (15.0) | 86.7 (12.0) | 80.8 (9.3) |

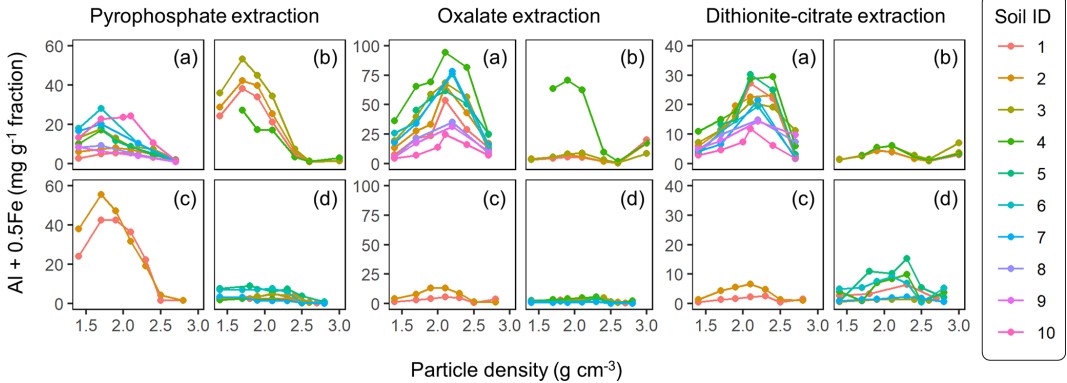

**Figure 1.** The concentrations of pedogenic metal (Al + 0.5 Fe) per fraction along the density gradient for allophanic Andisol group **(a)**, non-allophanic Andisol group **(b)**, Spodosol group **(c)**, and crystalline mineral group **(d)**. The metal extracted by initial pyrophosphate (left panels), subsequent acid oxalate (central panels), and final dithionite–citrate reagents (right panels) are shown. Each symbol represents an individual soil sample. Sample number (1–10) on the right corresponds to sample ID for each soil group in Table 1.

also showed a similar pattern to Fe in line with the generally positive correlation between extractable Fe and Mn (data not shown). The extractable Si showed less clear patterns along the density gradient (Fig. A4).

The Al : Fe molar ratio was highest in the metal$_{PP}$ phase (4.7 ± 4.3, mean ± SD), followed by the metal$_{OX}$ phase (2.6 ± 1.9) and, expectedly, lowest in the metal$_{DC}$ phase (0.4 ± 0.3) across all soils and their fractions. Several patterns were identified (Fig. A5). First, relative enrichment of Fe in metal$_{OX}$ and metal$_{DC}$ phases was evident in the highest-density fractions due presumably to the dissolution of crystalline Fe oxides. Second, from the low to meso density up to 2.3 g cm$^{-3}$, the Al : Fe ratio remained relatively constant with some exceptions. The allophanic Andisol group showed relative Al enrichment at around 1.8 g cm$^{-3}$ for the PP-extractable phase and at around 2.0–2.3 g cm$^{-3}$ for the metal$_{OX}$ phase. The non-allophanic Andisol group showed the Al enrichment towards low density for all extractable phases. For the other soil groups, similar Al enrichment was found in some samples, while others showed constant Al : Fe ratios. Nevertheless, the density at which the metal concentration was highest among the fractions (i.e., peak density) was quite similar between Fe and Al (Table A1). We thus examined the extractable Fe and Al together (i.e., Al + 0.5 Fe) for most of the subsequent analyses.

The extractable Al in most soils and fractions was more enriched relative to the corresponding extractable Si. The Al : Si molar ratio was 9.6 ± 16.9 (mean ± SD) for metal$_{PP}$, 3.6 ± 2.1 for metal$_{OX}$, and 1.1 ± 0.9 for the metal$_{DC}$ phase. Concerning density-dependent patterns, the Al : Si ratio in PP- and OX-extractable phases was higher at lower-density fractions for the non-allophanic Andisol and Spodosol groups, whereas a weak opposing trend was shown in the DC-extractable phase in the crystalline mineralogy group (Table A1).

We further compared the peak location in the unimodal metal concentration patterns along the density gradient among the studied soils (Figs. 1, A1 left panels). While the peak densities for all extractable metals were found at a meso-density range for all soil samples studied, we found important differences among the extractable phases. The peak densities of the metal$_{PP}$ phase were more variable compared to the metal$_{OX}$ and metal$_{DC}$ phases (Fig. 2). In particular, two C-rich soils (C-3, C-7) had the fitted peak density of metal$_{PP}$ at $< 1.5$ g cm$^{-3}$ (Fig. A1). The medians (and lower/upper quartiles) of the measured peak density among the 23 soil samples for PP-, OX- and DC-extractable metal phases were 1.7 (1.7–2.0), 2.1 (2.1–2.3), and 2.2 (2.1–2.3) g cm$^{-3}$, respectively (Fig. 2). Similarly, their means were 1.8, 2.2, and 2.2 g cm$^{-3}$. The peak densities estimated by the nor-

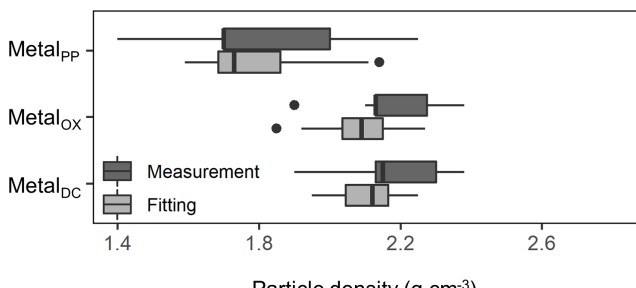

**Figure 2.** Boxplot showing the median and variation of densities at which the concentration of pedogenic metal ($Al + 0.5 Fe$) was the highest across the 23 studied soils. The peak density was determined for all three extractions (PP, OX, and DC) for each soil based on measurements (light color) and normal distribution fitting (dark color).

mal distribution fitting also showed comparable values. Thus, the peak densities of all three extractable phases were $< 2.3 \, \text{g cm}^{-3}$. The $metal_{PP}$ phase showed its peak consistently lower by $0.3–0.4 \, \text{g cm}^{-3}$ than the $metal_{OX}$ and $metal_{DC}$ phases (Figs. 1 and 2).

The concentrations of C co-dissolved during the initial PP and subsequent OX extractions ($DOC_{PP}$ and $DOC_{OX}$) correlated positively with the extractable metal concentrations (Fig. 3a–f). Coefficients of determination were highest for $Al_{PP}+0.5Fe_{PP}$ ($r^2 = 0.75$, $p < 0.0001$) followed by $Al_{PP}$ ($0.66$, $< 0.0001$) and then $Fe_{PP}$ ($0.56$, $< 0.0001$). Importantly, the positive $DOC_{PP}$–$Al_{PP}$ and/or $DOC_{PP}$–$Fe_{PP}$ relationships persisted across the density fractions within each soil, although the correlation was not significant for 5 out of the 23 soil samples (Fig. A6). The $DOC_{PP} : Al_{PP}$ mass ratio (i.e., slope of the regression lines) ranged from 2.9 to 28.2 (mean: 7.9, SE: 1.6), which is equivalent to the mean molar ratio of 17.8 (SE: 14.6).

Similarly, the organic matter co-dissolved by the oxalate extraction ($DOC_{OX}$) was positively related to $metal_{OX}$ owing largely to the allophanic Andisol samples (Fig. 3d–f). Simple linear regression of $DOC_{OX}$ against $Al_{OX}$, $Fe_{OX}$, and $Al_{OX} + 0.5 Fe_{OX}$ concentrations among all samples showed the strongest control by $Al_{OX}$ ($r^2 = 0.66$) followed by $Al_{OX} + 0.5 Fe_{OX}$ ($r^2 = 0.60$) and then $Fe_{OX}$ ($r^2 = 0.39$, $p < 0.0001$). When assessed for individual soils, the positive C–metal relationship (mostly only with $Al_{OX}$) persisted for 8 out of the 14 Andisol samples, but no positive relationship was present for the rest of the soils (Fig. A7). The range for the $DOC_{OX} : Al_{OX}$ mass ratio of the eight samples was 0.13–0.34 (molar $DOC_{OX} : Al_{OX}$ ratio of 0.29–0.76), and that of $DOC_{OX} : (Al_{OX} + 0.5 Fe_{OX})$ was 0.11–0.22 (molar $DOC_{OX} : Al_{OX} + Fe_{OX}$ ratio of 0.26–0.49) for these Andisol samples (data not shown). These ratios were roughly 10–50-fold lower than that from the PP-extractable phase.

The proportion of total C in each density fraction co-dissolved by initial PP and subsequent OX extractions (ex-

pressed as the sum of PP- and OX-extractable C) showed an increasing trend with increasing particle density (Fig. 4). Despite high variability especially towards higher-density fractions, the increasing trend with density was fitted by a polynomial curve for each of the four soil groups separately (Table A2). The three soil groups with high extractable metals tended to have higher proportions of extractable C compared to the crystalline mineralogy group. The allophanic Andisol group, which was characterized by higher $Al_{OX}$ and $Fe_{OX}$ concentrations, showed that appreciable amounts of OM were co-dissolved by the dissolution of $metal_{OX}$ phase (Fig. 4a). In contrast, the non-allophanic Andisols and Spodosol groups showed that nearly all the extractable C was released by the initial PP extraction. The non-allophanic Andisol group with characteristically high $Al_{PP}$ and $Fe_{PP}$ concentrations showed that a quarter up to nearly all of the C present in the higher-density fractions was co-dissolved by the initial PP extraction (Fig. 4b).

### 3.3 Distribution of mass, organic matter and extractable metal phases along the density gradient

The density fraction which accounted for the largest portion of bulk soil mass was in the $2.2–2.6 \, \text{g cm}^{-3}$ density range, and its median (and lower-upper quartiles) was 2.5 (2.2–2.6) $\text{g cm}^{-3}$ among the 23 soil samples (Fig. 5). Carbon distribution, calculated by multiplying the C concentration by the fractional mass for each density fraction, showed its peak in significantly lower yet still meso-density range (Fig. 5) with the median of 2.1 (1.9–2.2) $\text{g cm}^{-3}$. The data points at $1.4 \, \text{g cm}^{-3}$ were from three samples (A-1, S-1, and C-7) with very high total C values due to high OM input or reduced decomposition under cooler climate.

The extractable metals were also mainly concentrated in the meso-density range (Figs. A1 left, 5b), which is statistically indistinguishable from C peak as a whole sample set ($n = 23$). The medians (and lower–upper quartiles) of the peak density for the PP-, OX-, and DC-extractable phases were 2.1 (2.1–2.2), 2.1 (2.1–2.3), and 2.2 (2.1–2.4) $\text{g cm}^{-3}$, respectively. In contrast with the concentration-based patterns (Fig. 2), no clear difference was found between PP and the other two extractions due to the small mass contribution of the lower-density fractions where PP-extractable metal concentration was higher (Fig. A1a).

### 3.4 SEM observation of meso-density fractions

A clear shift in dominant particle type from plant detritus (POM) in the lowest-density fraction to aggregated particles in meso-density fractions, and finally to coarse mineral grains in the highest-density fraction, was observed for the three selected soils, one from the Spodosol and two from the crystalline mineralogy group (Fig. 6). The size of these particles ranged from a few tens to hundreds of micrometers in diameter. Similar density-dependent changes were previ-

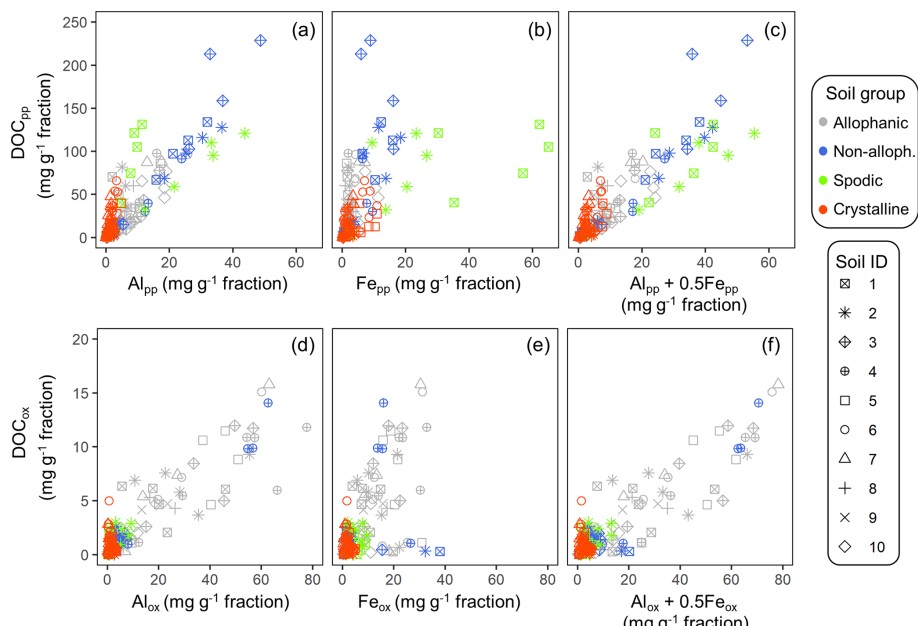

**Figure 3.** Scatter plot of extractable metals and co-dissolved organic C (DOC) for the density fractions from the studied soils. Pyrophosphate-extractable Al (**a**), Fe (**b**), and Al + 0.5 Fe (**c**) against $DOC_{OX}$ in the upper panel. Oxalate-extractable Al (**d**), Fe (**e**), and Al + 0.5 Fe (**f**) against $DOC_{OX}$ in the lower panel. Symbol color distinguishes the four soil groups and its shape corresponds to sample ID in Table 1.

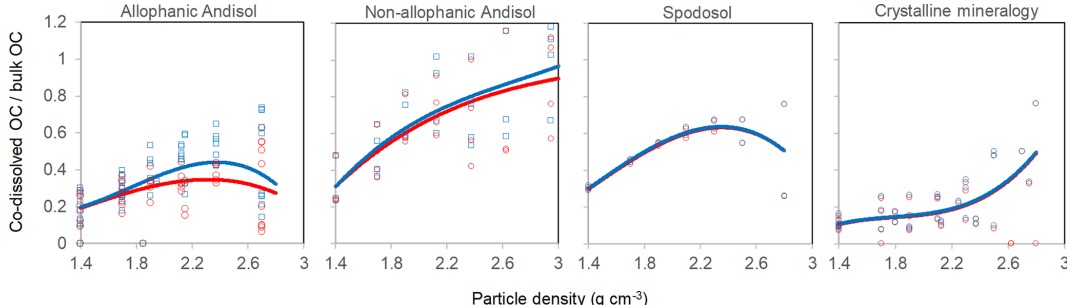

**Figure 4.** Proportions of bulk C in each density fraction co-dissolved by initial pyrophosphate extraction alone (red circle) and combined with subsequent acid oxalate extraction (blue rectangle), with polynomial fitting curves, for each soil group.

ously observed for one of the allophanic Andisols (A-3, Table 1, Wagai et al., 2014 TS5). Four density fractions (1.8–2.6 g cm$^{-3}$) where the majority of metals and OM reside were assessed by SEM in detail. In all three soils, 1.8–2.0 and 2.0–2.2 g cm$^{-3}$ fractions were more abundant in fragmented POM, which was mostly enmeshed in aggregates or coated with clay-size grains (Fig. 6a, b, e, f, i, j), while the materials in the 2.2–2.4 g cm$^{-3}$ fraction appeared largely aggregated with no visible POM (Fig. 6c, g, k). At a closer look at the surface of these aggregates and POM, clay-platelet-like features ($< 5\,\mu$m) were visible (Fig. 6a–k, magnified views). The next heavier fraction (2.4–2.6 g cm$^{-3}$) was more abundant in coarser mineral grains with clean surfaces, although some grains in this fraction were aggregated or showed rough surfaces (Fig. 6d, h, l, magnified views).

## 4  Discussion

### 4.1  Fe and Al phases extracted by the three reagents

The metals released by PP, OX, and DC extractions only *roughly* correspond to specific metal phases present in the soil as these extractions are not highly selective (Parfitt, 2009; Rennert, 2019). This approach, nevertheless, remains important as the extractable metal contents often show significant correlation with soil C storage and turnover times (Masiello et al., 2004; Percival et al., 2000; Porras et al., 2017; Torn et al., 1997; Wada and Higashi, 1976). It is thus critical to elucidate the nature of these metal phases including their localization, which aids in resolving the gap between commonly used mathematical models and the current understanding of soil C dynamics (Blankinship et al., 2018).

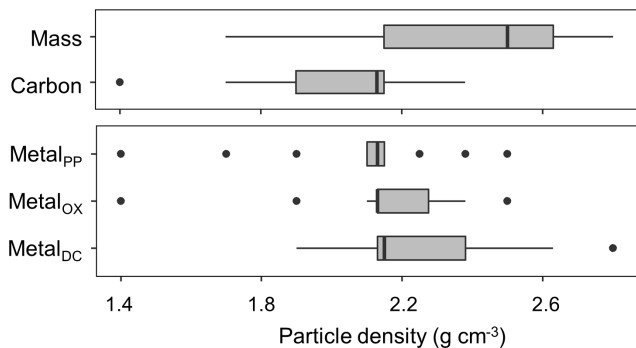

**Figure 5.** Boxplot showing the median and distribution of the densities at which the distribution of mass, C, and pedogenic metal (Al + 0.5 Fe) extracted by the three extractants (PP, OX, and DC) were the highest among the 23 studied soils.

Compared to common single extraction, a sequential extraction approach may allow better assignments of different Fe and Al phases (Dai et al., 2011; Shang and Tiessen, 1998). It is generally assumed that the PP extraction mainly dissolves organo-metal complexes (Bascomb, 1968; Takahashi and Dahlgren, 2016), whereas the OX and DC extractions target dissolution of short-range-order minerals and crystalline iron oxides, respectively (e.g., Inagaki et al., 2020; Lawrence et al., 2015; Shang and Tiessen, 1998). Pyrophosphate extraction data require particularly cautious interpretation due to the OM dissolution by high alkalinity (pH 10) and the dispersion or dissolution of colloidal and low-crystallinity Fe and Al oxide phases (Coward et al., 2018; Kaiser and Zech, 1996; Lawrence et al., 2015; Schuppli et al., 1983; Shang and Tiessen, 1998; Wagai et al., 2013). However, the significant $DOC_{PP}$–$metal_{PP}$ (especially $DOC_{PP}$–$Al_{PP}$) correlations found across the soils (Fig. 3a–c) and, to a limited extent, among the fractions within soils (Fig. A6), imply the predominance of strong OM–metal association phases such as organo-metal complexes in PP extracts because the potential artifacts, if occurring significantly, would have prevented the emergence of such a proportional relationship. Most of the studied soils showed significant C–metal correlation with high $DOC_{PP}$ : $Al_{PP}$ molar ratios (mean ± SE: 17.8±3.5, range: 6.6–63.3) and $DOC_{PP}$ : $Fe_{PP}$ ratios (75.6±18.0, 17.6–168, Fig. A6), in agreement with previous studies (Heckman et al., 2018; Wagai et al., 2013). While C : metal ratios of synthesized organo-metal associations vary widely depending on experimental conditions, higher ratios indicate the dominance of organo-metal complexes over adsorptive association with metal oxides (Wagai and Mayer, 2007). In laboratory coprecipitation experiments, the C : Fe molar ratio exceeding one led to organic encapsulation of Fe oxide particles (Kleber et al., 2015). Takahashi and Dahlgren (2016) estimated the C : metal molar ratio of 8.3 for organo-metal complexes in Andisols. We thus regard the PP-extractable phase as a mixture consisting largely of organo-metal complexes and their

coprecipitates with varying amounts of alkali-soluble or desorbable OM and non-centrifugeable colloidal Fe/Al oxide phases.

The OX-extractable metal phase is more likely influenced by short-range-order minerals (Parfitt and Childs, 1988 TS6; Rennert, 2019). We found strong positive correlation between $Al_{OX}$ and $Si_{OX}$ ($r^2 = 0.76$–$0.99$) with a relatively constant slope: the $Al_{OX}$ : $Si_{OX}$ molar ratio was 2.05 (allophanic Andisols), 2.25 (non-allophanic Andisols), 3.58 (Spodosols), and 3.91 (crystalline mineralogy group). Short-range-order aluminosilicates commonly found in Andisols and Spodosols have a molar ratio of 1–2 but possibly up to 4 for Al-rich allophane (Dahlgren et al., 1993). The OX-extractable phase in the studied soil fractions may also contain poorly crystalline gibbsite which can form rapidly in an OM-rich, acidic soil environment (e.g., Heckman et al., 2013). The source of $Al_{OX}$ and $Si_{OX}$ in the crystalline mineralogy group is less clear but likely to include interlayer components of 2 : 1 clay such as hydroxy Al polymers and aluminosilicates (Barnhisel and Bertsch, 1989; Wada and Kakuto, 1983) as well as amorphous gibbsite and silica (Drees et al., 1989). Most $Fe_{OX}$ phase is attributable to ferrihydrite and colloidal goethite for lower-density fractions and less-crystalline Fe oxides as well as magnetite – a primary mineral that associates little with OM due to the lack of hydroxylated surface – for higher-density fractions – for higher-density fractions (Cornell and Schwertmann, 2003; Parfitt and Childs, 1988 TS7; Rennert, 2019). The DC-extractable metal phase obtained after PP and OX extractions largely represents crystalline iron oxides and coprecipitated Al phases (Cornell and Schwertmann, 2003).

The sequential extraction results along the particle density gradient showed that greater proportions of total extractable Fe and Al were present as low-crystallinity phases (e.g., organic complexes, short-range-order minerals) in lower-density fractions (Fig. A4), which agrees well with the high affinity of these reactive phases to complex, coprecipitate, and adsorb OM (Kaiser and Guggenberger, 2003; Kleber et al., 2015; Wagai and Mayer, 2007; Wagai et al., 2013). The Al : Fe molar ratio of the extracts was relatively constant along the particle density gradient in some of the soils (Fig. A5), implying coprecipitation of the organo-Fe and organo-Al phases. The other soils, mostly in the Andisol and Spodosol groups, showed density-dependent patterns. Some of the allophanic Andisol samples, all of the non-allophanic Andisol samples, and one of the two Spodosol samples showed higher Al : Fe ratios towards lower-density fractions, especially in the $metal_{PP}$ and $metal_{OX}$ phases (Fig. A5a–c). These higher Al : Fe ratios may be explained by higher capability of Al ions to form an insoluble complex with organic ligands under low-pH and low metal : C ratio conditions (Nierop et al., 2002). In addition, the lower density of organo-Al coprecipitates ($1.7 \, \text{g cm}^{-3}$) than that of organo-Fe ones ($2.5 \, \text{g cm}^{-3}$, Kaiser and Guggenberger, 2007) may account for the observed higher Al : Fe ratios.

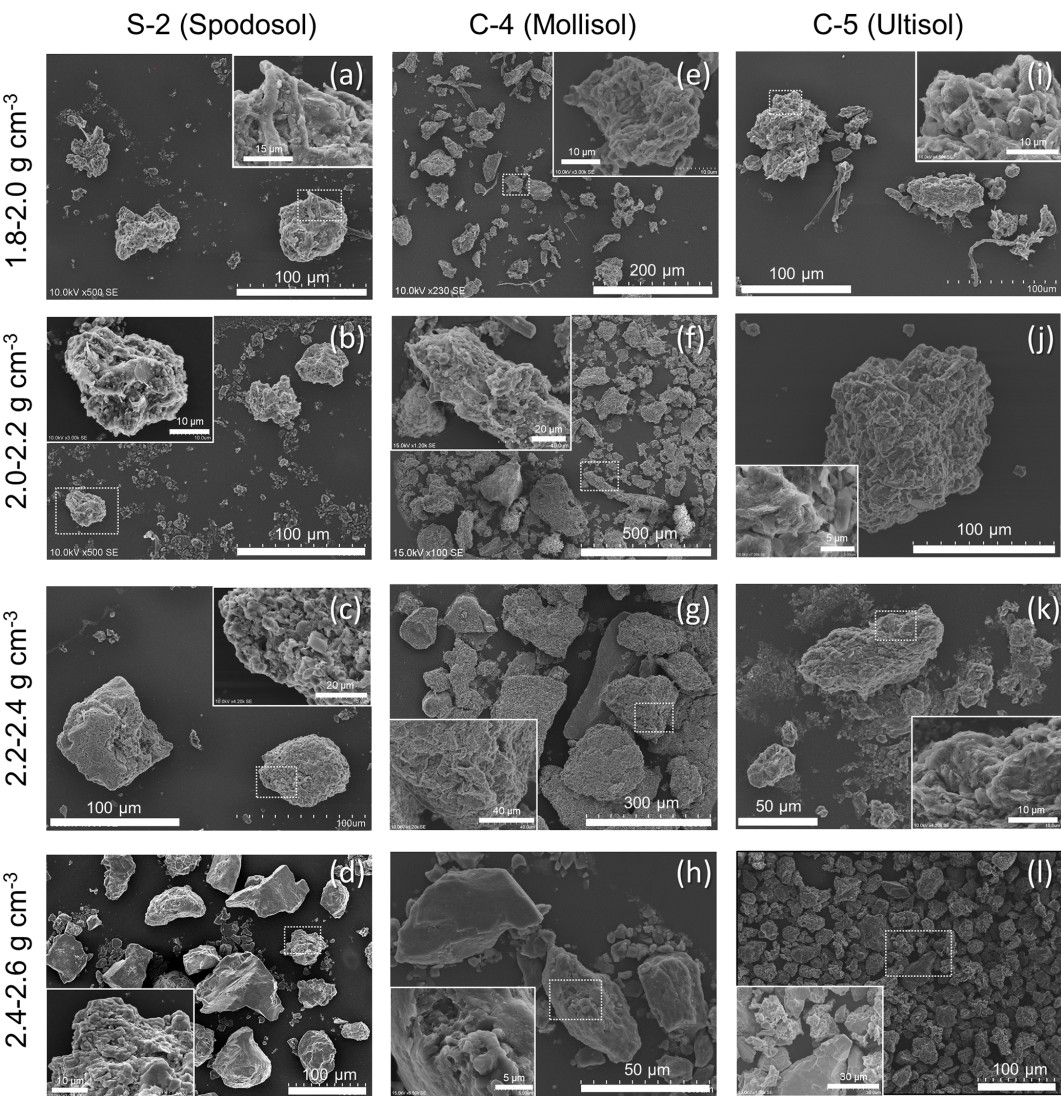

**Figure 6.** SEM images of three meso-density fractions and the adjacent higher-density fraction from S-2 soil (spodic horizon, **a–d**), C-4 soil (mollic horizon, **e–h**), and C-5 soil (kaolinitic A horizon, **i–l**). The enlarged view of the dotted rectangular section is shown at a corner, showing that the surface of selected particles was often aggregated or coated with finer materials such as clay platelets.

## 4.2   Pedogenic metal enrichment at meso-density range

The concentrations of extractable metal phases peaked at meso (intermediate) densities along the soil particle density gradient (Figs. 2, A1 left panels). Their dominance below the density cutoff of $2.4 \, \mathrm{g \, cm^{-3}}$ is explained only by their association with OM, which has a much lower density ($\sim 1.4 \, \mathrm{g \, cm^{-3}}$), supporting our hypothesis. While the cutoff densities are somewhat arbitrary, we define the range between 1.8 and $2.4 \, \mathrm{g \, cm^{-3}}$ as "meso density" for the following reasons: (i) a strong decline in OM concentration and C : N ratio above $\sim 1.8 \, \mathrm{g \, cm^{-3}}$ (Fig. A3a, b) suggests a major shift in OM source from plant detritus to microbially altered compounds as shown previously (e.g., Baisden et al., 2002), (ii) both the concentrations and distributions of the ex-

tractable metals began to increase at $> 1.8 \, \mathrm{g \, cm^{-3}}$ (Fig. A1), and (iii) most soil minerals have density $> 2.4 \, \mathrm{g \, cm^{-3}}$ (see Introduction). Thus, the meso-density fractions are characterized by enrichment of pedogenic Fe and Al phases and their association with microbially altered OM. The extractable metal phases in the meso densities made up less than one-fifth of soil masses, and the rest consisted largely of other minerals (e.g., crystalline clays) to form microaggregates resistant to mechanical shaking (Fig. 6). The meso-density enrichment of the metals thus implies their preferential association with OM relative to the other minerals.

Within the meso-density range, we found clear localization of different metal phases. The peak density of metal$_{\mathrm{PP}}$ concentration had a median of 1.8 (1.7–2.0) $\mathrm{g \, cm^{-3}}$, which was lower by 0.3–0.4 $\mathrm{g \, cm^{-3}}$ (on average) relative to

metal$_{OX}$ and metal$_{DC}$ across the soils (Figs. 2, A1 left). This difference remained the same when assessing Fe and Al separately. The lower peak density of metal$_{PP}$ can result either from the inherent low density of this phase as indicated by the high C : metal ratios (see Sect. 4.1, Fig. 3a–c) or from its attachment to low-density particles. The latter implies that the metal$_{PP}$ phase was preferentially associated with lower-density particles such as clay-covered POM (Fig. 6a, c), which would account for the presence of Fe$_{PP}$ phase despite its higher density than the Al$_{PP}$ phase.

Progressive changes in the concentrations of both organic and inorganic (mineral) phases along the density gradient (Figs. 1, A3a) are depicted for an idealized soil (Fig. 7a). While distinguishing between plant-derived POM in low-density fraction and mineral-associated OM (MAOM) in high-density fraction within bulk soil is a critical first step (Lavallee et al., 2020; Sollins et al., 1999), the transition from POM to MAOM is rather continuous, and the latter contains a wide array of OM–mineral associations (Hatton et al., 2012; Jones and Singh, 2014; Sollins et al., 2009; Turchenek and Oades, 1979; Wagai et al., 2018), as conceptualized in the "soil continuum model" of soil OM formation (Lehmann and Kleber, 2015). The higher-density fractions (e.g., > 2.4 g cm$^{-3}$) are increasingly dominated with primary minerals and Fe-bearing minerals, including crystalline Fe oxides (Jones and Singh, 2014; Sollins et al., 2009) with small amounts of N-rich OM (Fig. A3b). The lower-density fractions, on the other hand, hold increasing amounts of POM with appreciable levels of PP-extractable organo-metal phases, especially at around the 1.8 g cm$^{-3}$ range (Figs. 1, 2). In the meso-density fractions (1.8–2.4 g cm$^{-3}$), where the major portions of OM and extractable metals were located, significant portions (20 % up to nearly 100 %) of the OM therein were co-dissolved by PP and OX extractions (Fig. 4). By applying a similar sequential extraction method to four soil profiles of contrasting mineralogy, Heckman et al. (2018) reported, on average, that 70 % of total C was extractable. This extractable OM possibly existed in metal-bound forms (Coward et al., 2017; Wagai et al., 2013; Zhao et al., 2016). The remaining, non-extractable OM in the meso-density fraction is presumably stabilized by mechanisms other than simple adsorptive association with the extractable metal phases.

### 4.3 Organo-metal association, aggregation, and OM stabilization

Organo-metal associations take place at multiple spatial scales within a bulk soil. Organo-metallic complexation, sorption, and coprecipitation occur at molecular to colloidal scales (Kleber et al., 2015). Current density fractionation results, on the other hand, suggest that the extractable metals and associated OM contribute to the formation of meso-density aggregates with a few tens to hundreds of micrometers in diameter (Fig. 6). At the aggregate scale, organo-

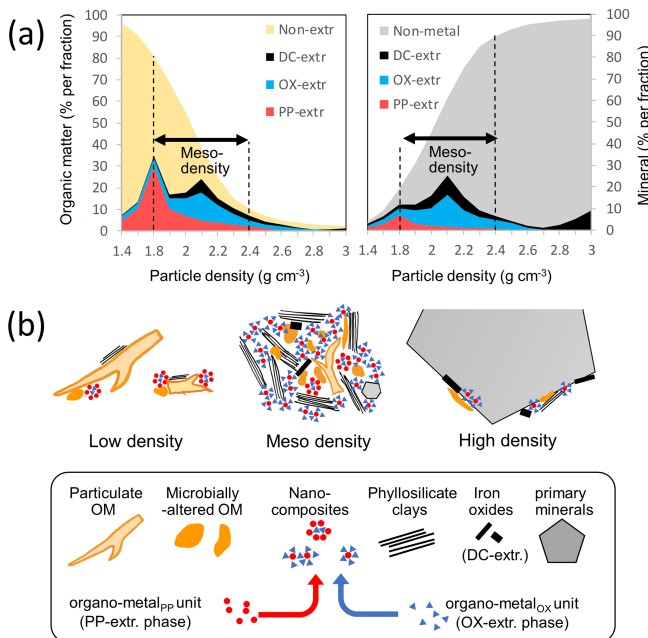

**Figure 7. (a)** Changes in the concentration of organic and mineral phases along soil particle gradient for an idealized soil. The concentrations of OM (left plot) and metals (right plot) extracted sequentially by pyrophosphate (PP), acid oxalate (OX), and dithionite (DC) per density fraction were shown in different colors. Non-extractable OM includes both particulate and microbially altered OM. Non-metal mineral phase includes phyllosilicate clays and primary minerals. **(b)** Schematic representation of low-, meso-, and high-density particles (upper) and their building blocks (lower). PP- and OX-extractable phases were presumed to be present as "nanocomposites" that act as glue to form meso-density microaggregates. The distribution of the nanocomposites across density fractions can explain the C : metal proportional relationship found in Figs. 3, A6, and A7. The nanocomposites rich in organo-metal$_{PP}$ unit are more abundant in OM-rich environments (e.g., lower-density fraction) relative to those rich in organo-metal$_{OX}$ units. Objects do not reflect the size difference among them.

mineral interactions occur with a much higher level of complexity (Keil and Mayer, 2014; Totsche et al., 2017). Here, we discuss how colloidal-scale interaction of metals and OM may be linked with micron-scale aggregate formation to account for the observed density-dependent patterns.

The observed proportionality between extractable metal phases and associated OM among the density fractions gives some hints to bridge between the colloidal and larger-scale associations. Significant C–metal correlations in PP- and OX-extractable phases, previously found across a range of bulk soils (Wagai et al., 2013), were shown among the density fractions (Fig. 3) and even within each soil in many cases (Figs. A6, A7). Specifically, positive DOC$_{PP}$–metal$_{PP}$ correlation (largely DOC$_{PP}$–Al$_{PP}$) was found for all soils except for three samples from the crystalline mineralogy group (Fig. A6), while such a correlation for the subsequent

OX extraction was limited to 8 out of the 14 Andisol samples (Fig. A7). Relatively constant C : metal ratios across the density fractions imply that the organo-metal association formed in the field remained intact as a physical unit during the fractionation steps, and these units were distributed among the fractions. These organo-metal units themselves are presumably present as colloid-sized "nanocomposites" (Fig. 7b) consisting of precipitated organo-metal complexes (i.e., metal$_{PP}$ phase) and, at least in the case of Andisols, OM-sorbed metal oxides (i.e., metal$_{OX}$ phase), such as those identified in soils using high-resolution imaging techniques (Asano et al., 2018; Wen et al., 2014). Another feature is that these nanocomposites must be attached to larger particles containing some combination of low-density OM, high-density mineral, or more of each other. Without sufficient size, Stokes' law predicts that they would have remained in density liquid and been CE1 lost during the centrifugation step.

We hypothesize that these organo-metal-rich nanocomposites function as a glue or effective binding agent (Asano et al., 2018) and promote ternary associations of OM, metal, and clays (Wagai and Mayer, 2007), as depicted in Fig. 7b. In fact, the meso-density materials were largely present as microaggregates with abundant clays on their surfaces (Fig. 6). For the two soil samples (A-3, C-4) that we further size-fractionated following the density fractionation, 59 %–84 % of the mass in the meso-density fractions consisted of < 2 μm sized particles (isolated after maximum dispersion by sonication) that were enriched in the extractable metals relative to bulk samples (unpublished data), in support of our hypothesis. How do these nanocomposites form and function? The Al and Fe are trivalent metals and can act as (monomeric or polynuclear) glue between different organic ligands, particles, and surfaces. An organic particle or coating may stick to a mineral surface via van der Waals interactions but become aggregated to other organic particles via a polyvalent metal connection. Furthermore, monomeric Fe and Al can form various ternary complexes in the presence of OM and other dissolved inorganic species found in soil solution such as Ca and Si, thereby preventing their polymerization (Adhikari et al., 2019; Tamrat et al., 2019; Yang et al., 2017). These organo-metal-rich mixed-phase nanocomposites, acting as glues (organo-metallic glue hypothesis), can give a mechanistic explanation for the moderately strong C–metal correlations among the density fractions (Figs. 3, A6, A7) as well as for the dominance of OM, metal, and presumably clay in the meso-density range (Figs. 5, A1 right panels).

Such micro-scale aggregation can enhance OM stability by reducing the accessibility of microbes, exo-enzymes, and/or e-acceptors (Balesdent et al., 2000; Keil and Mayer, 2014; Lehmann et al., 2007; Sollins et al., 1996). Among the pedogenic metal phases, crystalline Fe oxides (roughly corresponds to the metal$_{DC}$ phase) strongly enhance microaggregation, particularly in highly weathered soils (e.g., Shang and Tiessen, 1998). This metal phase can protect relatively small amounts of OM for a prolonged time (e.g., Eusterhues et al., 2003; Mikutta et al., 2006). Short-range-order mineral could also contribute to aggregation and thus physical protection of OM within such a mineral matrix. Microaggregates in Andisols (especially metal$_{OX}$-rich ones) show high physical stability (Shoji et al., 1993) even against wet oxidation and reductive dissolution treatments (Churchman and Tate, 1986). Stable ternary associations of OM, low-crystallinity Fe oxide, and microporous allophane in an Andisol has been hypothesized (Filimonova et al., 2016). A portion of soil Fe phase such as low-crystallinity Fe oxyhydroxide, and presumably associated OM, can survive harsh dithionite extraction (Coward et al., 2018), in some cases due to the protective effect of metal$_{OX}$ phases. Even after strong dispersion by sonication (up to $1500\,\mathrm{J\,mL^{-1}}$), 60 %–70 % of total C and extractable metals in Andisols remained in the meso-density fractions (Basile-Doelsch et al., 2007; Wagai et al., 2015) that were largely present as micron- and submicron-sized aggregates (Asano and Wagai, 2014; Asano et al., 2018). Similarly, the main C storage location in tropical Ferrasols was sonication-resistant particles that were characterized by slightly higher density ($2.45$–$2.8\,\mathrm{g\,cm^{-3}}$), enrichment of halloysite, and resistance to ∼ 200 years of cultivation (Basile-Doelsch et al., 2009). Compared to these metal oxide phases, organo-metal complexes are more labile, for instance, with a change in pH (Takahashi and Dahlgren, 2016). Thus their contribution to aggregation may be lower, although this phase can be physically occluded within stable microaggregates. Faster turnover (more $^{14}$C enrichment) of PP-extractable C compared to the C associated with other mineral phases (Heckman et al., 2018) as well as the metal$_{PP}$ enrichment at lower density (Figs. 1, 2) support this view. While the relative importance of specific metal phases remains to be elucidated, these extractable metals likely contribute to OM stabilization by promoting aggregation via organo-metallic glues as well as by direct organo-metal interaction via complexation and adsorption.

## 4.4 Co-localization of metal and microbially altered OM at meso-density fraction

We further considered the distributions of metal and OM along the density gradient to translate the observed results into field-level processes. Their distributions are determined by two variables: mass distribution and the concentration of the respective elements. The peak density of mass distribution was quite variable among the studied soils (Fig. 5), which will be further examined in our companion study (Kajiura et al., 2020 TS8). Consequently, the peak densities of metal phases were also moderately variable. The mean peak densities of both C and all metal phases were, however, ∼ $2.1\,\mathrm{g\,cm^{-3}}$ (Fig. 5). In fact, the meso-density range ($1.8$–$2.4\,\mathrm{g\,cm^{-3}}$) accounted for $59\pm14\,\%$ and $64\pm15\,\%$ (mean ± SD) of total C and N, respectively, among the studied soils. Similarly, more than half of the total extractable

metals were in the meso-density range ($65 \pm 17$ % of $Fe_{PP}$, $63 \pm 15$ % of $Al_{PP}$, $56 \pm 14$ % of $Fe_{OX}$, $70 \pm 13$ % of $Al_{OX}$, $61 \pm 17$ % of $Fe_{DC}$, and $66 \pm 15$ % of $Al_{DC}$). The meso-density co-localization of OM and the extractable metals found here thus suggest that metal binding via multiple physicochemical processes discussed above contributes to the stability of pedogenic metal phases against leaching/dissolution and that of OM against biological degradation. Our view is consistent with C and N isotope tracer studies that identified meso-density fractions as the main reservoir of stabilized C across a wide range of soils (Baisden et al., 2002; Crow et al., 2014; Hatton et al., 2012; Sollins et al., 2009; Wagai et al., 2018; Jones and Singh, 2014) with an exception of Fe-rich weathered soils where $> 2.4\,\mathrm{g\,cm^{-3}}$ fractions store more C due to the abundance of heavy minerals (Jones and Singh, 2014; Sollins et al., 2009).

The co-localization of OM and metals is illustrated for the three forest soils developed through contrasting pedogenesis (Fig. 8). In the tropical Ultisol, the DC-extractable phase accounted for greater proportions of total extractable metal than the other two soils (Fig. 8a). Accordingly, this phase explained the majority of the OM extractable by PP, OX, and DC (Wagai et al., 2013). The spodic horizon sample under cooler climate stored more C in lower-density fractions and held major portions of metals as PP-extractable phase (Fig. 8b), in agreement with the podzolization concept (Lundström et al., 2000). The allophanic Andisol held much higher amounts of extractable metals and C than the other two soils (Fig. 8c). Co-localization at a narrower density range in the Andisol can be attributable to higher contents of organo-metallic glue.

## 4.5   Implications

The similar distributions of OM and extractable Fe and Al found here (Figs. 5, 8, A1) imply a common set of processes that promote the formation of organo-metal associations across a range of pedogenic environments. Almost all samples showed a unimodal distribution with the peaks in the meso-density range (Figs. 1, 2, and A1). The three processes operating at a fundamental level are (1) the production of microbially altered OM from the original low-density plant detritus, (2) the release of metals from high-density, weatherable minerals, and (3) the formation of organo-metal-rich nanocomposites and concurrent incorporation of low- and high-density materials (most importantly, clays) into meso density via their gluing properties – the organo-metallic glue hypothesis (Fig. 8).

From the organic side, the most consistent change along the particle density gradient was the progressive decline in C : N ratio and OM concentration from low- to high-density fractions (Fig. A3), which is generally explained by the shift from POM to microbially processed compounds (Baisden et al., 2002; Gunina and Kuzyakov, 2014). Microbially driven, oxidative depolymerization increases the sol-

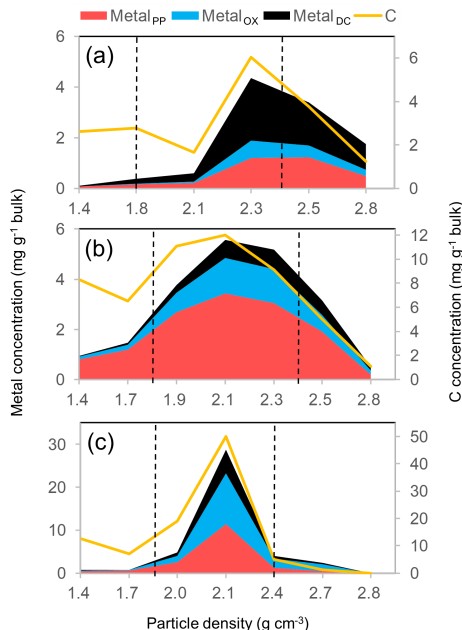

**Figure 8.** The distribution of the extractable metal phases ($Al + 0.5\,Fe$) on the left $y$ axis and that of C on the right $y$ axis along the density gradient for three undisturbed forest soils. A kaolinitic Ultisol (O-5 in Table 1) developed from sedimentary rock under tropical wet climate **(a)**, a Spodosol Bhs horizon (S-2) with quartz and plagioclase-rich mineralogy from glacial till under cool temperate moist climate **(b)**, and an allophanic Andisol (A-10) from tephra under warm temperate moist climate **(c)**. Vertical dotted lines show the meso-density range.

ubility, the number of ionized functional groups (especially carboxylic groups), and acidity thereby enhancing the reactivity of remaining OM with metals and mineral surfaces (Heckman et al., 2013; Kleber et al., 2015). The accumulation of the pedogenic metal phases from low to meso-density fractions (Figs. 2, A1) thus suggests that oxidative depolymerization of POM appears to be necessary for the organo-metal associations. From the mineral side, metals released by weathering can readily bind to organic ligands. For instance, negatively charged bacterial cell surface attracts metal cations, which leads to the nucleation and precipitation of low-crystallinity Fe oxyhydroxides and aluminosilicates (Ferris et al., 1989; Urrutia and Beveridge, 1994). Tamrat et al. (2019) showed nano-sized Fe-Al-Si coprecipitate formation from biotite weathering solution in the presence of low-molecular organic acid (termed nanosized coprecipitates of inorganic oligomers with organics, "nanoCLICs"). These reactions likely promote the formation of the nanocomposites having a relatively narrow range of OM : metal ratios that can act as relatively persistent glue to bind soil particles, most importantly phyllosilicate clays that themselves strongly bind with OM.

The three processes identified (Fig. 9) are ultimately driven by the factors driving pedogenesis – physicochemi-

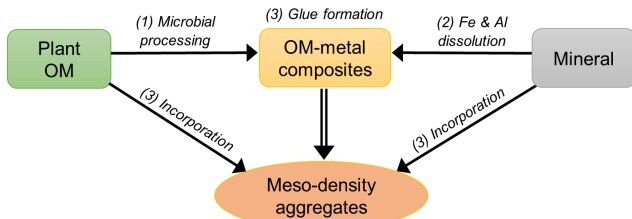

**Figure 9.** Conceptual illustration of meso-density aggregate formation. The three main processes are (1) microbial processing of plant-derived organic matter, (2) metal dissolution via chemical weathering, and (3) the formation of organo-metallic glue, which promote meso-density microaggregate formation by incorporating some POM and mineral.

cal forces (heat, water, acidity) and biological activity. Then the environmental conditions promoting the three processes in balance would lead to local maxima of the organo-metallic glue and meso-density aggregates (see Fig. A8 and the discussion therein). Such condition at the global scale includes acidic soils under wetter climate (Rasmussen et al., 2018) especially with the parent materials that are abundant in weatherable minerals. At the pedon scale, the interface of O/A horizons or the B horizons that experience podzolization would fit with this condition. At smaller scales, micro spots having redox fluctuation and rhizosphere likely promote the organo-metallic glue formation due to the abundance of organic ligands and active dissolution/precipitation of Fe and other mineral phases (Chen et al., 2020; Garcia Arredondo et al., 2019; Keiluweit et al., 2015; Yu, 2018). The current view and the growing evidence on rapid formation of various organo-metal-mineral associations at submicron scale (e.g., Basile-Doelsch et al., 2015 TS9; Garcia Arredondo et al., 2019; Heckman et al., 2013) suggest that the concept of soil C saturation and soil's capacity to protect C based on clay and silt contents (e.g., Six et al., 2002) require refinement.

Our results and proposed hypothesis may help to integrate some of the important findings and concepts in the literature. The predominance of stabilized OM in organo-mineral fractions (e.g., $< 20\,\mu m$ size class or meso-density range) has been shown by physical fractionation studies (e.g., Christensen, 2001; Six et al., 2000; von Lützow et al., 2007), but the involvement of pedogenic metal phases was much less studied. Protective effects of phyllosilicate clays (e.g., Barré et al., 2014) and pedogenic metal phases (e.g., Porras et al., 2017; Wagai and Mayer, 2007) on OM are likely to occur concurrently, and possibly synergistically, within meso-density microaggregates. Enrichment of certain clay minerals in meso-density fractions (e.g., smectite in a Vertisol, kaolinite in Spodosol and Oxisol, Jones and Singh, 2014) and enhanced physical stability of clay aggregates by goethite particle incorporation (Dultz et al., 2019) support this idea. The aggregate hierarchy concept recognized

the role of low-crystallinity mineral phases and microbial compounds as persistent binding agents (Tisdall and Oades, 1982 TS10) but not their interaction or formation pathways. With new analytical techniques and methodologies, our understanding of microaggregate formation (e.g., Asano and Wagai, 2014; Lehmann et al., 2007; Totsche et al., 2017) and molecular-scale interaction of OM, metals, and other inorganic phases (e.g., Chen et al., 2020; Tamrat et al., 2019) is advancing. We believe that a remaining key question is how these molecular-scale interactions are related to soil physical fractions and the formation of a hierarchical aggregate structure. The current study provides some insights to this end. Further efforts to fill the scale gap will be important to better understand soil's protective capacity to store OM and for the development of more mechanistic biogeochemical models.

**Appendix A**

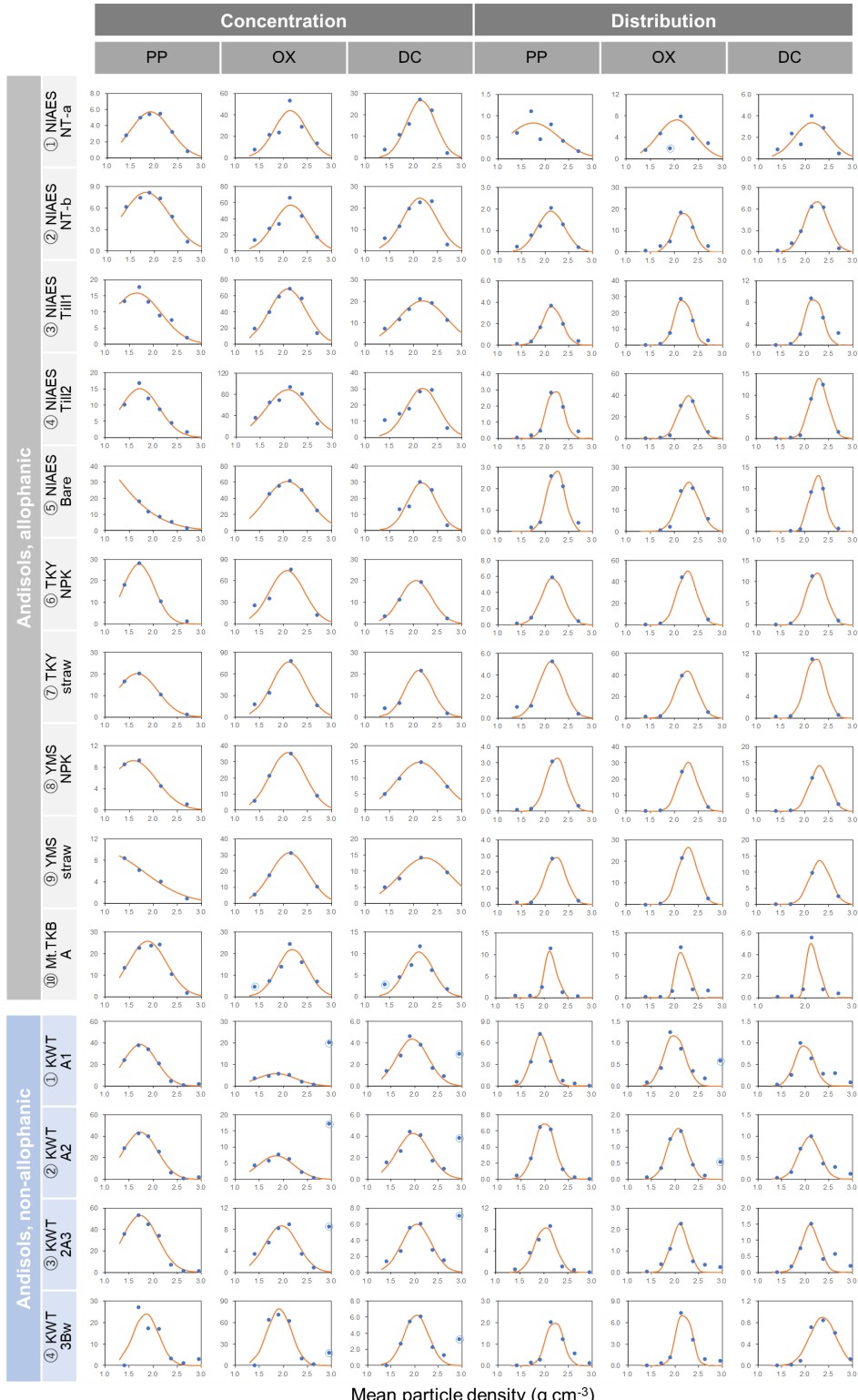

**Figure A1.**

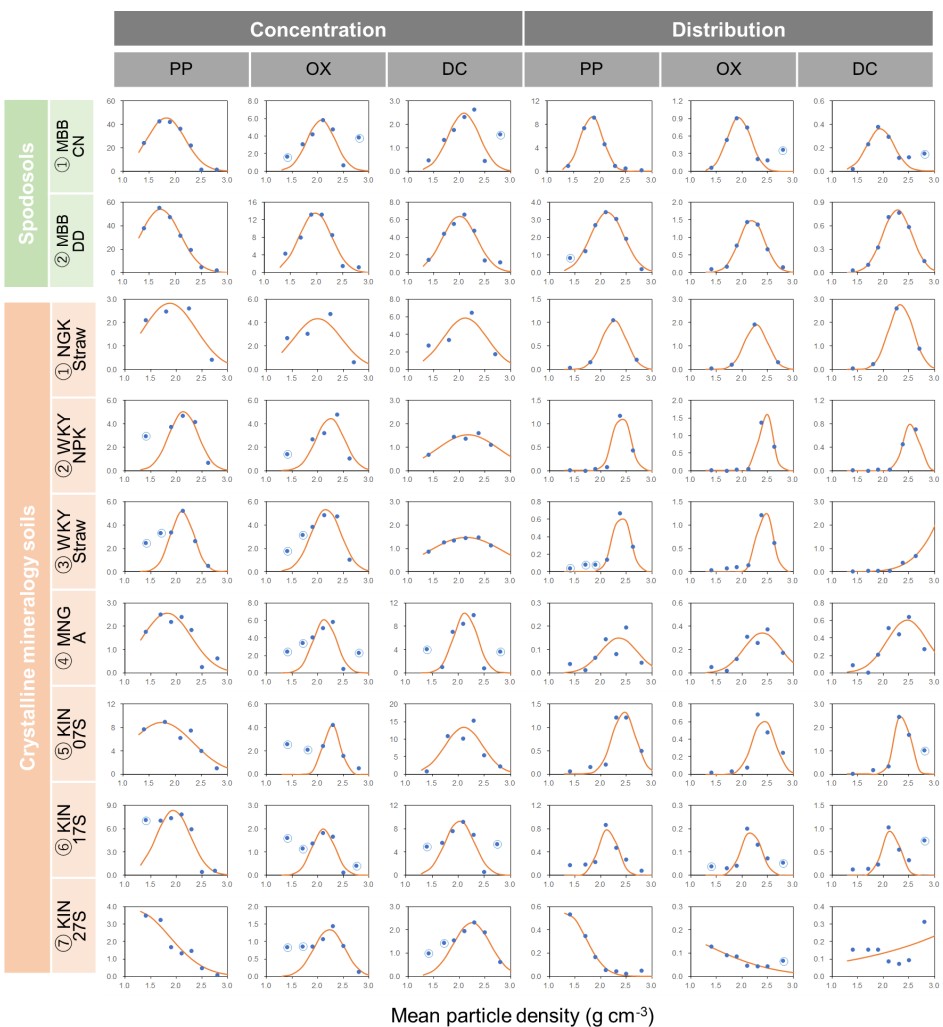

**Figure A1.** Extractable metal (Al + 0.5 Fe) against mean particle density for each soil sample. The metal concentrations (mg g$^{-1}$ fraction) from pyrophosphate (PP), acid oxalate (OX), and dithionite–citrate (DC) extractions are shown in the left three panels. The distributions of PP, OX, and DC-extractable metal phases (mg g$^{-1}$ whole soil) are in the right three panels. Normal distribution curve fit is shown as line. The data points circled in blue were omitted from the fitting.

https://doi.org/10.5194/soil-6-1-2020

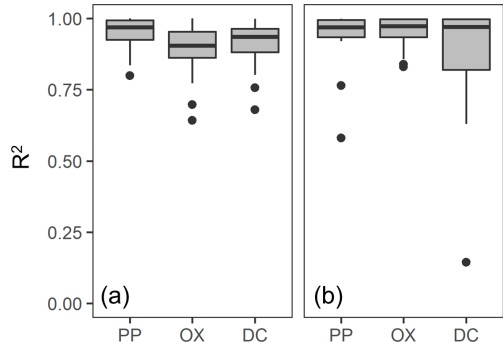

**Figure A2.** Boxplot of $r^2$ value for the normal distribution curve fitting of the extractable metal concentration **(a)** and distribution **(b)** against mean particle density. The fitting was done for each soil and each extraction as shown in Fig. A1.

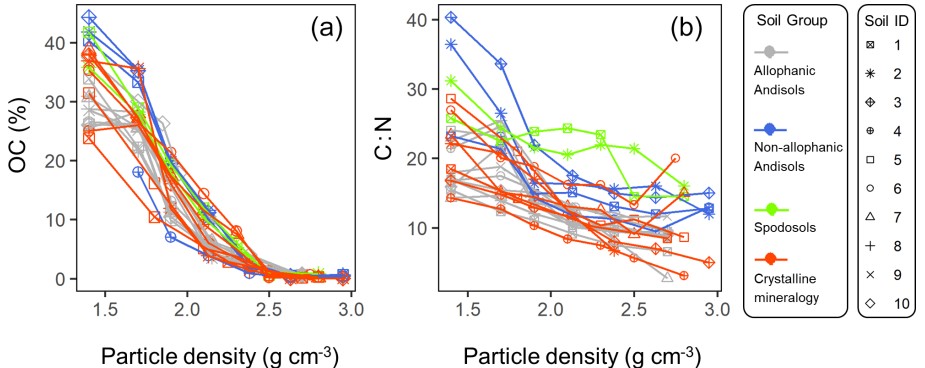

**Figure A3.** Line graphs showing organic C concentration per fraction, as weight % **(a)** and C : N ratio **(b)** against soil particle density. Each line represents each soil sample which belongs to one of the four soil groups. Soil ID numbers, shown in different symbol shapes, correspond to Table 1. TS11

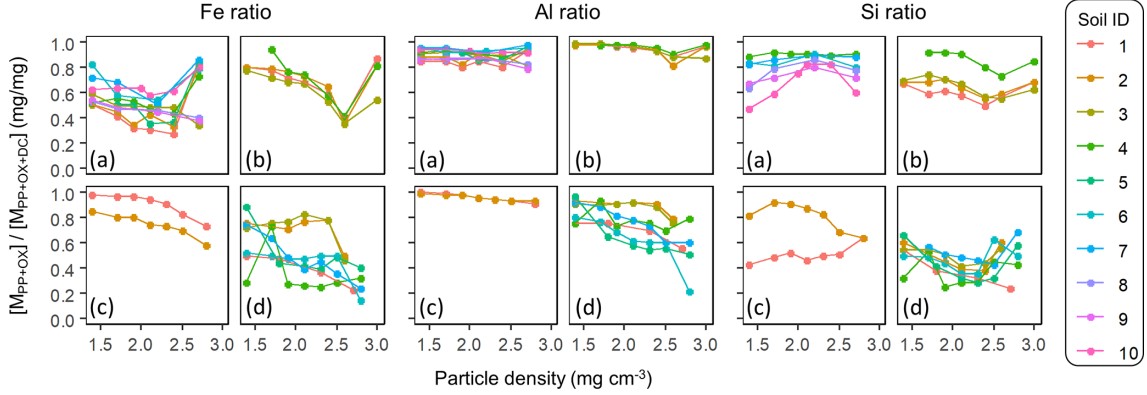

**Figure A4.** Proportion of total extractable Fe (left), Al (center), and Si (right panels) present as pyrophosphate- plus oxalate-extractable phases. This ratio obtained from each density fraction was shown along the density gradient for allophanic Andisols **(a)**, non-allophanic Andisols **(b)**, Spodosols **(c)**, and crystalline mineralogy group **(d)**. Each symbol represents the individual soil sample ID in Table 1.

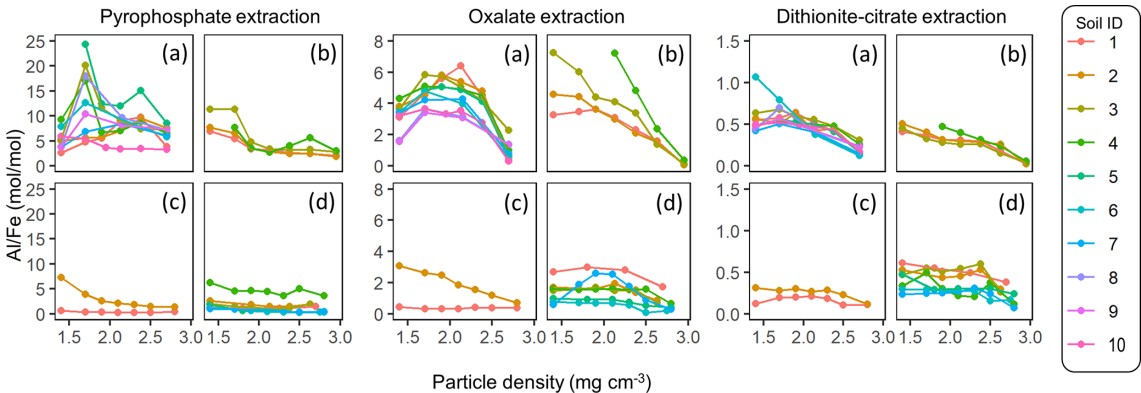

**Figure A5.** Molar Al-to-Fe ratio of each extractable phase from each density fraction along the density gradient. The ratio was shown for initial pyrophosphate (left four plots) and subsequent acid oxalate (middle four plots), and final dithionite–citrate extractions (right four plots) were shown for allophanic Andisols **(a)**, non-allophanic Andisols **(b)**, Spodosols **(c)**, and crystalline mineralogy group **(d)**. Each symbol represents the individual soil sample ID in Table 1.

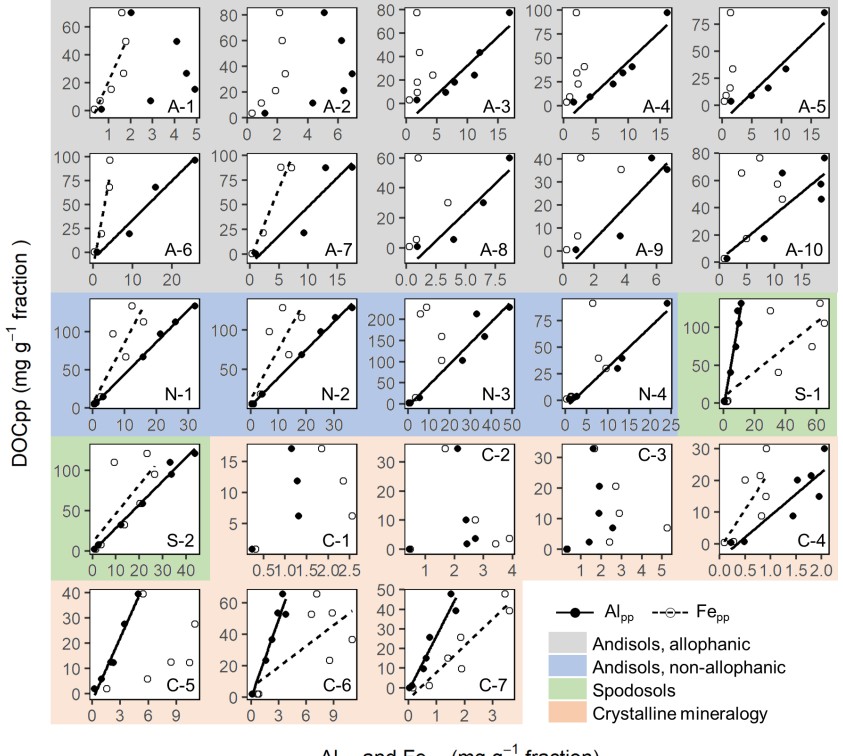

**Figure A6.** Scattered plots of pyrophosphate-extractable Fe and Al against dissolved organic C concentration for each soil sample. Soil sample ID in each plot corresponds to that from Table 1. Solid lines represent significant linear regressions at $p < 0.1$.

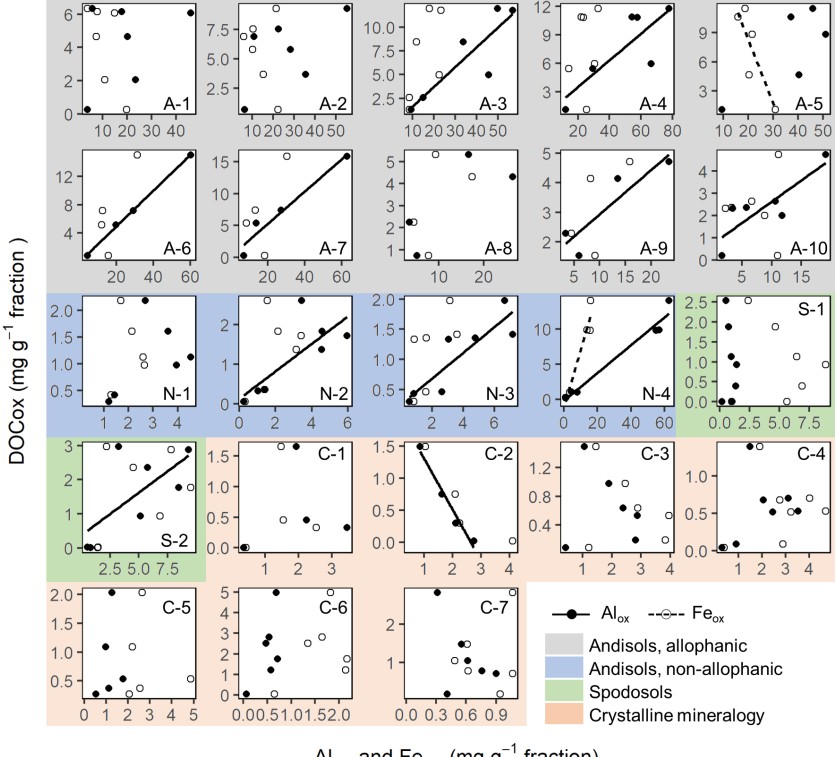

**Figure A7.** Scattered plots of acid-oxalate-extractable Fe and Al against dissolved organic C concentration for each soil sample. Soil sample ID in each plot corresponds to that from Table 1. Solid lines represent significant linear regressions at $p < 0.1$.

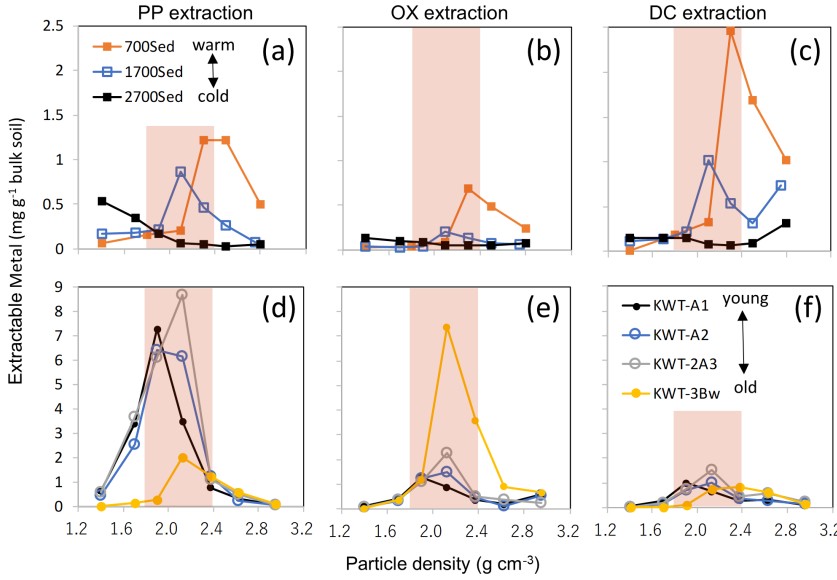

**Figure A8.** The distribution of the extractable metal phases ($Al + 0.5\,Fe$) along the particle density gradient for two soil series. The upper three plots show PP- **(a)**, OX- **(b)**, and DC-extractable metals **(c)** for three A horizon soils under tropical forest along an elevation gradient from a weathered soil under warm climate at 700 m (sample ID: C5), moderately weathered soil at 1700 m (C6), and much less weathered soil at 2700 m (C7). The lower panels **(d, e, f)** showed surface and buried horizons of a non-allophanic Andisol profile (sample ID: N1–N4) that received multiple tephra deposits. Shaded zones show the meso-density range.

Discussion associated with Fig. A8

The relative balance of the three processes (OM supply and microbial processing, metal dissolution via weathering, and their binding/aggregation, Fig. 9) appear to control the nature of meso-density metal enrichment. Environmental gradients (e.g., climo- and chrono-sequences) give clear shifts in the balance of the three. Along a strong temperature gradient on a forested mountain slope, surface soils at higher altitudes experience slower rates of OM decay (and thus higher OM accumulation) and mineral weathering than those at warmer lower-altitude soils. The majority of pedogenic metals in the 2700 m soil altitude (C-5) was present as PP-extractable phase and their distribution peaked at $< 2.0\,\mathrm{g\,cm^{-3}}$ fractions, whereas the 700 m altitude soil (C-7) held the metals mostly as DC- and PP-extractable phases that were mainly present at $> 2.2\,\mathrm{g\,cm^{-3}}$, and the mid-altitude soil (C-6) showed the patterns in between them (Fig. A8a–c). Meso-density metal enrichment was strongest in the 700 m soil. This result can be explained by (1) higher microbial processing of OM as the forest productivity, and thus OM supply to soil was the highest (Kitayama and Aiba, 2002; Wagai et al., 2008), and (2) higher degree of weathering under the warm moist climate regime (Wagai et al., 2009).

Along with a volcanic soil profile experiencing up-building pedogenesis, we also observed the decline in metal$_{PP}$ and a concurrent increase in metal$_{OX}$ over pedogenic time (from upper to lower horizons), implying a shift in the dominant metal phase from organo-metal complexes to short-range-order mineral. Concurrently, we found the increase in metal distribution from low towards higher density (Fig. A8d–f). The top three horizons (A1, A2, and 2A3) were developed over the last ca. 10 kyr based on key tephra (Hijiori pumice) identified between the 2A3 and 3Bw horizons, suggesting that the organo-metallic glue enriched in the PP-extractable phase was replaced by that rich in the OX-extractable phase over ca. 10 kyr under a reduced OM input (buried) condition.

These results therefore suggest general patterns in the localization of specific metal phases and the mode of OM–metal association across scales as follows: (i) Al and Fe released by weathering preferentially bind to organic ligands to form an organo-metallic complex in OM-rich environment (e.g., surface horizon especially under cooler climate) and (ii), under the pedogenic condition where net organic supply to mineral surface is limited for a prolonged time (e.g., rapid decomposition under a warm, wet condition and deeper horizons), polymerization of Fe and Al (and Si) is facilitated. The latter leads to the formation of short-range-order mineral which is further transformed to well-crystalline oxides over time. In other words, pedogenic reactions such as dynamics of mobile weathering products and secondary mineral formation occurring at a pedon scale (e.g., the depth-dependent framework, Lawrence et al., 2015) can be observed within bulk soils by the current approach combining sequential density fractionation with the extraction of operationally defined metal phases.

**Table A1.** Concentrations of OC, N, and the elements extracted by pyrophosphate, acid-oxalate, and dithionite–citrate reagents in bulk and density fractions from each of 23 soil samples. TS1 2

| Soil ID | Mean density (g cm⁻³) | Low cutoff (g cm⁻³) | High cutoff (g cm⁻³) | Mass fraction TS1 | OC (mg g⁻¹) | N (mg g⁻¹) | C/N (mg g⁻¹) | PP Al (mg g⁻¹) | PP Fe (mg g⁻¹) | PP Si (mg g⁻¹) | PP Mn (mg g⁻¹) | PP DOC (mg g⁻¹) | PP TDN (mg g⁻¹) | OX Al (mg g⁻¹) | OX Fe (mg g⁻¹) | OX Si (mg g⁻¹) | OX Mn (mg g⁻¹) | OX TDN (mg g⁻¹) | DC Al (mg g⁻¹) | DC Fe (mg g⁻¹) | DC Si (mg g⁻¹) | DC Mn (mg g⁻¹) | PP-extr Al+0.5Fe (mg g⁻¹) | OX-extr Al+0.5Fe (mg g⁻¹) | DC-extr Fe (mg g⁻¹) |
|---|---|---|---|---|---|---|---|---|---|---|---|---|---|---|---|---|---|---|---|---|---|---|---|---|---|
| **A-1/ NIA-NTa** | 1.40 | | 1.60 | 0.22 | 339 | 21.9 | 15.5 | 2.02 | 1.60 | NA | 0.29 | 70.21 | 6.64 | 5.76 | 3.81 | 3.05 | 0.13 | 0.41 | 1.40 | 5.28 | 3.52 | 0.07 | 2.82 | 7.66 | 4.04 |
| | 1.70 | 1.60 | 1.80 | 0.22 | 222 | 18.0 | 12.4 | 4.10 | 1.79 | NA | 0.39 | 49.41 | 2.89 | 17.69 | 7.82 | 9.70 | 0.28 | 0.50 | 3.87 | 13.84 | 5.39 | 0.07 | 4.99 | 21.60 | 10.80 |
| | 1.90 | 1.80 | 2.00 | 0.09 | 121 | 10.9 | 11.1 | 4.54 | 1.69 | NA | 0.34 | 26.57 | 2.03 | 20.08 | 7.43 | 10.29 | 0.25 | 0.42 | 6.09 | 19.72 | 5.35 | 0.16 | 5.39 | 23.80 | 15.95 |
| | 2.13 | 2.00 | 2.25 | 0.15 | 47 | 5.1 | 9.2 | 4.92 | 1.13 | NA | 0.18 | 15.16 | 1.26 | 46.02 | 14.87 | 13.42 | 0.48 | 0.66 | 6.31 | 31.63 | 7.18 | 0.20 | 5.48 | 53.45 | 27.25 |
| | 2.38 | 2.25 | 2.50 | 0.13 | 22 | 2.9 | 7.5 | 2.92 | 0.62 | NA | 0.11 | 7.09 | 0.79 | 23.33 | 10.90 | 2.88 | 0.37 | 0.28 | 6.51 | 26.21 | 4.71 | 0.04 | 3.23 | 28.79 | 22.12 |
| | 2.70 | 2.50 | | 0.22 | 4.6 | 0.75 | 6.2 | 0.68 | 0.37 | NA | 0.03 | 0.93 | 0.23 | 3.72 | 19.75 | 0.37 | 0.17 | 0.04 | 0.33 | 4.05 | 1.16 | 0.04 | 0.86 | 13.59 | 2.35 |
| | BULK | | | | 149 | 10.0 | 14.9 | 1.60 | 1.50 | 1.16 | 0.37 | NA | NA | 19.10 | 8.60 | 11.11 | 0.34 | NA | 3.60 | 15.10 | 4.29 | 0.10 | 2.35 | 23.40 | 11.15 |
| **A-2/ NIA-NTb** | 1.40 | | 1.60 | 0.04 | 309 | 18.2 | 17.0 | 5.09 | 2.14 | NA | 0.49 | 81.68 | 3.36 | 10.73 | 5.82 | 5.65 | 0.23 | 0.41 | 2.11 | 7.70 | 3.54 | 0.06 | 6.16 | 13.64 | 5.96 |
| | 1.70 | 1.60 | 1.80 | 0.10 | 218 | 15.7 | 13.9 | 6.23 | 2.32 | NA | 0.48 | 59.74 | 3.13 | 22.55 | 10.27 | 12.37 | 0.35 | 0.54 | 3.78 | 15.36 | 4.17 | 0.08 | 7.39 | 27.68 | 11.46 |
| | 1.90 | 1.80 | 2.00 | 0.15 | 113 | 9.4 | 12.0 | 6.92 | 2.51 | NA | 0.33 | 34.28 | 2.42 | 28.25 | 10.09 | 13.96 | 0.37 | 0.48 | 7.45 | 24.28 | 4.76 | 0.21 | 8.17 | 33.29 | 19.59 |
| | 2.13 | 2.00 | 2.25 | 0.28 | 58 | 5.5 | 10.6 | 6.38 | 1.88 | NA | 0.17 | 20.89 | 1.60 | 55.42 | 21.35 | 32.73 | 0.56 | 0.89 | 6.51 | 32.09 | 3.62 | 0.15 | 7.32 | 66.09 | 22.55 |
| | 2.38 | 2.25 | 2.50 | 0.27 | 26 | 2.8 | 9.5 | 4.34 | 0.94 | NA | 0.11 | 11.45 | 0.85 | 35.51 | 15.31 | 19.24 | 0.25 | 0.39 | 7.07 | 32.41 | 4.09 | 0.23 | 4.81 | 43.17 | 23.27 |
| | 2.70 | 2.50 | | 0.17 | 5.7 | 0.65 | 8.9 | 1.18 | 0.32 | NA | 0.04 | 3.59 | 0.28 | 6.19 | 22.26 | 4.10 | 0.15 | 0.07 | 0.43 | 5.45 | 0.72 | 0.04 | 1.34 | 17.32 | 3.15 |
| | BULK | | | | 80 | 5.9 | 13.6 | 3.50 | 1.20 | 0.47 | 0.18 | NA | NA | 35.50 | 14.00 | 18.94 | 0.41 | NA | 5.60 | 26.60 | 3.49 | 0.18 | 4.10 | 42.50 | 18.90 |
| **A-3/ NIA-TillI** | 1.40 | | 1.60 | 0.01 | 261 | 16.4 | 15.9 | 11.12 | 4.33 | NA | 0.22 | 24.21 | 1.63 | 15.13 | 8.59 | 7.15 | 0.24 | 0.16 | 2.73 | 8.93 | 2.23 | 0.06 | 13.29 | 19.43 | 7.19 |
| | 1.70 | 1.60 | 1.80 | 0.02 | 262 | 12.0 | 21.9 | 16.89 | 1.74 | NA | 0.12 | 77.64 | 2.94 | 33.66 | 11.96 | 15.96 | 0.50 | 0.39 | 4.58 | 13.92 | 1.98 | 0.09 | 17.75 | 39.64 | 11.54 |
| | 1.90 | 1.80 | 2.00 | 0.13 | 105 | 8.0 | 13.1 | 11.98 | 2.13 | NA | 0.13 | 43.58 | 2.58 | 49.65 | 17.97 | 24.84 | 0.68 | 0.91 | 5.87 | 20.47 | 2.48 | 0.10 | 13.04 | 58.63 | 16.10 |
| | 2.13 | 2.00 | 2.25 | 0.42 | 56 | 5.2 | 10.8 | 7.87 | 1.79 | NA | 0.07 | 18.26 | 1.29 | 56.84 | 23.30 | 28.73 | 0.80 | 1.09 | 7.28 | 27.14 | 2.74 | 0.13 | 8.77 | 68.49 | 20.85 |
| | 2.38 | 2.25 | 2.50 | 0.27 | 23 | 2.4 | 9.6 | 6.47 | 1.82 | NA | 0.09 | 9.65 | 0.71 | 45.47 | 22.47 | 24.79 | 0.90 | 0.53 | 6.02 | 26.21 | 2.61 | 0.13 | 7.38 | 56.70 | 19.12 |
| | 2.70 | 2.50 | 0.20 | 7.1 | 0.77 | 0.44 | 1.73 | 0.49 | NA | 0.04 | 0.04 | 0.34 | NA | 8.66 | 4.73 | 0.37 | 0.15 | 2.59 | 17.38 | 1.17 | 0.10 | 1.97 | 13.79 | 11.28 | 6.07 |
| | BULK | | | | 51 | 4.1 | 12.5 | 5.00 | 1.00 | 0.47 | 0.05 | NA | NA | 41.60 | 18.50 | 21.06 | 0.71 | NA | 6.30 | 25.50 | 2.57 | 0.14 | 5.50 | 50.85 | 19.05 |
| **A-4/ NIA-TillII** | 1.40 | | 1.60 | 0.01 | 255 | 21.4 | 11.9 | 2.06 | 1.12 | 0.22 | 0.34 | 1.48 | 29.36 | 14.08 | 11.90 | 0.50 | 0.25 | 3.52 | 14.81 | 1.75 | 0.08 | 10.23 | 36.39 | 10.93 | 14.93 |
| | 1.70 | 1.60 | 1.80 | 0.01 | 283 | 11.5 | 24.6 | 16.04 | 1.95 | 1.14 | 0.11 | 97.47 | 3.34 | 54.35 | 22.12 | 21.54 | 0.92 | 0.44 | 5.03 | 19.80 | 1.94 | 0.12 | 17.02 | 65.41 | 17.89 |
| | 1.90 | 1.80 | 2.00 | 0.04 | 131 | 8.9 | 14.7 | 10.61 | 3.25 | 0.55 | 0.13 | 40.48 | 2.22 | 57.52 | 23.49 | 23.61 | 0.88 | 0.74 | 6.01 | 23.77 | 2.48 | 0.13 | 12.23 | 69.26 | 28.70 |
| | 2.13 | 2.00 | 2.25 | 0.32 | 65 | 6.3 | 10.3 | 7.68 | 2.26 | 0.15 | 0.08 | 22.65 | 1.20 | 77.76 | 32.86 | 33.41 | 1.07 | 1.14 | 9.33 | 38.75 | 3.44 | 0.24 | 8.81 | 94.18 | 29.55 |
| | 2.38 | 2.25 | 2.50 | 0.42 | 28 | 2.7 | 10.4 | 4.15 | 0.97 | BD | 0.05 | 8.94 | 0.48 | 66.27 | 30.44 | 29.41 | 1.35 | 0.58 | 9.30 | 40.52 | 3.44 | 0.25 | 4.63 | 81.49 | 29.55 |
| | 2.70 | 2.50 | 0.20 | 0.25 | 6.0 | 0.70 | 8.6 | 1.55 | 0.49 | BD | 0.02 | 3.32 | NA | 12.24 | 25.40 | 0.37 | 0.39 | 0.13 | 1.21 | 9.71 | 0.63 | 0.04 | 1.80 | 24.94 | 6.07 |
| | BULK | | | | 42 | 3.6 | 11.8 | 3.81 | 1.27 | 0.56 | 0.08 | 13.33 | 1.04 | 53.38 | 29.82 | 23.44 | 0.95 | NA | 6.49 | 31.39 | 2.73 | 0.17 | 4.44 | 68.29 | 22.18 |
| **A-5/ NIA-Bare** | 1.40 | | 1.60 | 0.01 | 262 | 10.9 | 24.1 | NA | NA | NA | NA | NA | NA | NA | NA | NA | NA | NA | NA | NA | NA | NA | NA | NA | NA |
| | 1.70 | 1.60 | 1.80 | 0.01 | 259 | 11.2 | 23.2 | 17.23 | 1.47 | NA | 0.10 | 86.04 | 3.21 | 37.22 | 15.85 | 18.00 | 0.46 | 0.46 | 4.69 | 17.20 | 2.02 | 0.10 | 17.96 | 45.14 | 13.29 |
| | 1.90 | 1.80 | 2.00 | 0.04 | 99 | 7.4 | 13.4 | 10.77 | 1.79 | NA | 0.12 | 33.66 | 2.23 | 45.90 | 18.75 | 26.30 | 0.53 | 0.86 | 4.96 | 19.77 | 2.23 | 0.10 | 11.66 | 55.28 | 14.85 |
| | 2.13 | 2.00 | 2.25 | 0.30 | 52 | 5.2 | 10.0 | 7.80 | 1.35 | NA | 0.10 | 16.00 | 1.40 | 50.96 | 21.58 | 26.30 | 0.50 | 0.88 | 9.85 | 40.69 | 4.19 | 0.31 | 8.47 | 61.75 | 30.19 |
| | 2.38 | 2.25 | 2.50 | 0.40 | 24 | 2.6 | 9.6 | 4.92 | 0.68 | NA | 0.05 | 9.07 | 0.70 | 40.39 | 20.32 | 20.87 | 0.80 | 0.49 | 7.08 | 35.90 | 2.89 | 0.19 | 5.26 | 50.55 | 25.03 |
| | 2.70 | 2.50 | | 0.24 | 6.3 | 0.69 | 9.2 | 1.48 | 0.36 | NA | 0.02 | 3.49 | 0.26 | 9.52 | 30.94 | 5.25 | 0.44 | 0.12 | 0.43 | 5.53 | 0.33 | 0.04 | 1.67 | 24.99 | 3.20 |
| | BULK | | | | 37 | 3.0 | 12.3 | 3.60 | 0.50 | 0.51 | 0.04 | NA | NA | 39.60 | 18.50 | 20.08 | 0.59 | NA | 5.80 | 30.10 | 2.40 | 0.15 | 3.85 | 48.85 | 20.85 |
| **A-6/ TKY-NPK** | 1.40 | | 1.60 | 0.01 | 265 | 15.8 | 16.8 | 15.84 | 4.18 | 2.57 | 0.32 | 68.01 | 5.90 | 19.74 | 11.92 | 10.84 | 0.28 | 0.31 | 1.79 | 3.47 | 2.52 | NA | 17.93 | 25.70 | 3.53 |
| | 1.70 | 1.60 | 1.80 | 0.03 | 262 | 15.0 | 17.5 | 25.86 | 4.23 | 2.40 | 0.24 | 96.26 | 4.41 | 29.02 | 12.51 | 15.33 | 0.57 | 0.41 | 4.78 | 12.46 | 3.95 | NA | 27.98 | 35.28 | 11.01 |
| | 2.15 | 1.80 | 2.50 | 0.58 | 59 | 4.5 | 13.0 | 9.19 | 2.05 | 0.87 | 0.08 | 19.22 | 0.98 | 60.23 | 31.27 | 39.33 | 0.94 | 1.16 | 5.19 | 28.51 | 4.18 | NA | 10.21 | 75.87 | 19.44 |
| | 2.70 | 2.50 | 0.38 | | 0.51 | 0.53 | 0.97 | 0.34 | 0.23 | 0.01 | 0.46 | BD | 4.27 | 0.25 | 4.14 | 0.17 | 0.08 | 0.24 | 4.16 | 1.08 | NA | 1.14 | 0.46 | 2.32 | |
| | BULK | | | | 46 | 3.9 | 11.8 | 8.09 | 1.69 | 0.79 | 0.10 | 16.52 | 0.99 | 42.97 | 25.57 | 28.30 | 0.75 | 0.87 | 3.03 | 19.46 | 2.79 | NA | 8.94 | 55.75 | 12.76 |
| **A-7/ TKY-Comp** | 1.40 | | 1.60 | 0.06 | 21.5 | 14.3 | 13.00 | 7.15 | 2.42 | 0.30 | 0.38 | 6.93 | 13.82 | 8.37 | 8.58 | 0.11 | 0.38 | 1.21 | 5.96 | 2.43 | NA | 16.58 | 18.01 | 4.19 | |
| | 1.70 | 1.60 | 1.80 | 0.06 | 256 | 17.5 | 14.6 | 17.60 | 5.35 | 2.08 | 0.37 | 87.77 | 6.15 | 27.27 | 13.34 | 16.56 | 0.53 | 0.51 | 2.15 | 8.83 | 3.15 | NA | 20.27 | 33.94 | 6.57 |
| | 2.15 | 1.80 | 2.50 | 0.51 | 63 | 5.1 | 12.4 | 9.28 | 2.28 | 0.94 | 0.15 | 21.52 | 1.13 | 63.04 | 30.44 | 40.34 | 1.01 | 1.28 | 6.08 | 31.06 | 4.95 | NA | 10.42 | 78.26 | 21.61 |
| | 2.70 | 2.50 | | 0.35 | 5.7 | 2.1 | 2.8 | 1.01 | 0.34 | 0.24 | 0.02 | 0.50 | BD | 6.91 | 18.23 | 0.89 | 0.18 | 0.11 | 0.21 | 3.00 | NA | 0.02 | 1.18 | 16.02 | 1.71 |
| | BULK | | | | 60 | 5.4 | 11.2 | 6.98 | 1.88 | 0.81 | 0.15 | 20.15 | 1.31 | 36.91 | 22.47 | 24.93 | 0.64 | 0.82 | 2.79 | 18.01 | 2.87 | NA | 7.92 | 48.15 | 11.80 |
| **A-8/ YMS-NPK** | 1.40 | | 1.60 | 0.01 | 288 | 12.8 | 22.5 | NA* | NA* | NA* | NA* | NA* | NA* | NA* | NA* | NA* | NA* | NA* | NA* | NA* | NA* | NA* | NA* | NA* | NA* |
| | 1.70 | 1.60 | 1.80 | 0.01 | 283 | 11.1 | 25.6 | 8.70 | 1.00 | 0.43 | 0.15 | 59.94 | 2.52 | 16.57 | 9.37 | 6.04 | 0.43 | 0.21 | 3.94 | 11.78 | 1.76 | 0.00 | 9.20 | 21.25 | 9.83 |
| | 2.15 | 1.80 | 2.50 | 0.70 | 37 | 3.5 | 10.5 | 4.01 | 0.87 | 0.70 | 0.05 | 5.68 | 0.39 | 26.40 | 17.37 | 13.57 | 0.83 | 0.40 | 4.29 | 21.24 | 2.26 | 0.13 | 4.45 | 35.09 | 14.90 |
| | 2.70 | 2.50 | | 0.29 | 7.1 | 0.77 | 9.3 | 0.96 | 0.28 | 0.27 | 0.01 | 0.72 | BD | 5.18 | 7.74 | 2.33 | 0.33 | 0.08 | 1.32 | 12.08 | 0.75 | 0.13 | 1.10 | 9.05 | 7.36 |
| | BULK | | | | 29 | 2.8 | 11.0 | 2.54 | 0.54 | 0.51 | 0.05 | 4.90 | 0.32 | 21.26 | 13.78 | 9.03 | NA | NA | 3.80 | 19.93 | 1.82 | NA | 2.80 | 28.16 | 13.77 |
| **A-9/ YMS-Straw** | 1.40 | | 1.60 | 0.02 | 286 | 15.9 | 17.9 | 6.65 | 3.72 | 2.83 | 0.18 | 35.44 | 1.62 | 3.39 | 4.48 | 1.30 | 0.04 | 0.13 | 1.59 | 6.98 | 2.03 | NA | 8.51 | 5.63 | 5.08 |
| | 1.70 | 1.60 | 1.80 | 0.02 | 252 | 13.4 | 18.8 | 5.69 | 1.13 | 1.39 | 0.34 | 40.42 | 1.34 | 13.47 | 8.20 | 5.27 | 0.39 | 0.22 | 2.74 | 10.00 | 2.67 | NA | 6.25 | 17.57 | 7.74 |
| | 2.15 | 1.80 | 2.50 | 0.69 | 35 | 3.0 | 11.7 | 3.65 | 0.93 | 0.88 | 0.11 | 6.49 | 0.44 | 23.49 | 15.81 | 10.21 | 0.78 | 0.41 | 4.16 | 20.37 | 2.79 | NA | 4.11 | 31.40 | 14.34 |
| | 2.70 | 2.50 | 0.27 | | 0.65 | 0.78 | 0.83 | 0.24 | 0.30 | 0.02 | 0.45 | BD | 5.97 | 9.14 | 2.58 | 0.38 | 0.13 | 1.81 | 15.76 | 1.14 | NA | 0.94 | 10.53 | 9.69 | |
| | BULK | | | | 35 | 3.2 | 11.0 | 2.26 | 0.56 | 0.67 | 0.10 | 6.68 | 0.42 | 19.09 | 12.95 | 8.36 | 0.60 | 0.37 | 3.83 | 19.94 | 2.51 | NA | 2.54 | 25.56 | 13.80 |

| Soil ID | Mean density $g\,cm^{-3}$ | Low cutoff $g\,cm^{-3}$ | High cutoff $g\,cm^{-3}$ | Mass fraction | OC $mg\,g^{-1}$ | N $mg\,g^{-1}$ | C/N $mg\,g^{-1}$ | PP Al | PP Fe | PP Si | PP Mn | PP DOC | PP TDN | OX Al | OX Fe | OX Si | OX Mn | OX TDN | DC Al | DC Fe | DC Si | DC Mn | PP-extr $mg\,g^{-1}$ | OX-extr Al+0.5Fe $mg\,g^{-1}$ | DC-extr $mg\,g^{-1}$ |
|---|---|---|---|---|---|---|---|---|---|---|---|---|---|---|---|---|---|---|---|---|---|---|---|---|---|
| A-10/ MTT-A | 1.40 | 1.60 | 1.60 | 0.03 | 376 | 22.2 | 17.0 | 11.38 | 4.01 | 1.15 | 0.44 | 65.48 | 4.17 | 3.42 | 2.21 | 0.96 | 0.08 | 0.14 | 0.91 | 3.82 | 2.35 | 0.03 | 13.38 | 4.53 | 2.82 |
|  | 1.70 | 1.60 | 1.80 | 0.02 | 300 | 19.5 | 15.4 | 19.08 | 7.27 | 1.65 | 0.45 | 76.58 | 4.93 | 5.74 | 3.25 | 1.72 | 0.13 | 0.15 | 1.54 | 6.15 | 2.39 | 0.04 | 22.71 | 7.37 | 4.62 |
|  | 1.85 | 1.80 | 1.90 | 0.01 | 263 | 17.3 | 15.2 | 17.3 | NA | 2.18 | 0.40 | 57.42 | 3.63 | 3.68 | 3.57 | 1.85 | 0.30 | 0.20 | 2.37 | 10.02 | 1.95 | 0.06 | 23.71 | 13.99 | 7.37 |
|  | 1.95 | 1.90 | 2.00 | 0.11 | 178 | 13.2 | 13.5 | 18.44 | 10.53 | 2.28 | 0.40 | 46.23 | 2.91 | 3.68 | 6.34 | 2.64 | 0.17 | 0.67 | 3.68 | 9.64 | 1.85 | 0.12 | 24.63 | 24.63 | 11.78 |
|  | 2.13 | 2.00 | 2.25 | 0.47 | 105 | 8.2 | 12.7 | 18.53 | 11.39 | 2.18 | 0.40 | 46.23 | 2.91 | 7.17 | 8.81 | 0.94 | 0.82 | 0.37 | 1.81 | 8.65 | 1.08 | 0.12 | 24.63 | 13.99 | 7.37 |
|  | 2.38 | 2.25 | 2.50 | 0.12 | 40 | 3.3 | 12.3 | 3.26 | 4.92 | 0.40 | 0.27 | 17.27 | 1.05 | 4.16 | 4.16 | 0.39 | 0.25 | 0.16 | 0.25 | 3.03 | 1.08 | 0.07 | 10.59 | 16.15 | 6.14 |
|  | 2.63 | 2.50 | 2.75 | 0.17 | 4.8 | 0.46 | 10.3 | 0.72 | 0.61 | 0.34 | 0.05 | 5.52 | 0.15 | 0.34 | 0.63 | 0.32 | 0.05 | 0.02 | 0.14 | 1.56 | 0.44 | 0.02 | 1.71 | 7.15 | 1.76 |
|  | 2.70 | 2.50 |  | 0.23 | 98 | 7.6 | 13.7 | 1.30 | 0.81 | 0.92 | 0.27 | 2.62 | 1.05 | 1.64 | 8.81 | 0.39 | 0.82 | 0.16 | 2.44 | 10.30 | 1.44 | 0.05 | 10.59 | 16.15 | 7.59 |
|  | BULK |  |  |  | 98 | 7.6 | 13.7 | 12.64 | 7.52 | 1.46 | NA | 37.08 | 2.43 | 12.27 | 7.72 | 0.99 | 0.53 | 0.22 | 2.44 | 10.30 | 1.44 | 0.05 | 15.35 | 2.22 | 3.88 |
| N-1/ KWT-A1 | 1.40 | 1.60 | 1.60 | 0.03 | 404 | 17.4 | 23.2 | 21.16 | 6.33 | 2.84 | 0.28 | 97.27 | 4.23 | 2.68 | 1.70 | 1.55 | 0.06 | 0.09 | 0.40 | 2.05 | 1.63 | 0.03 | 24.32 | 3.53 | 1.43 |
|  | 1.70 | 1.60 | 1.80 | 0.09 | 333 | 15.6 | 21.4 | 21.99 | 12.29 | 3.61 | 0.31 | 134.07 | 6.23 | 3.61 | 2.15 | 0.69 | 0.17 | 0.08 | 0.74 | 4.20 | 2.05 | 0.03 | 42.30 | 5.66 | 2.84 |
|  | 1.90 | 1.80 | 2.00 | 0.21 | 196 | 13.1 | 14.9 | 15.97 | 11.50 | 2.60 | 0.34 | 112.80 | 6.34 | 2.59 | 2.15 | 1.03 | 0.27 | 0.07 | 1.08 | 7.12 | 2.33 | 0.05 | 38.14 | 4.68 | 2.60 |
|  | 2.13 | 2.00 | 2.25 | 0.17 | 101 | 6.7 | 15.1 | 10.40 | 15.83 | 2.80 | 0.40 | 66.72 | 3.51 | 3.95 | 2.64 | 0.94 | 0.50 | 0.06 | 0.89 | 7.12 | 2.04 | 0.05 | 33.96 | 5.83 | 4.64 |
|  | 2.38 | 2.25 | 2.50 | 0.17 | 20 | 1.5 | 13.3 | 3.26 | 2.55 | 1.85 | 0.19 | 14.48 | 0.56 | 1.44 | 3.16 | 0.39 | 0.46 | 0.07 | 2.62 | 5.95 | 2.04 | 0.04 | 21.03 | 5.27 | 3.87 |
|  | 2.63 | 2.50 | 2.75 | 0.32 | 4.8 | 0.52 | 9.2 | 0.61 | 0.34 | 0.03 | NA | 5.52 | NA | 0.34 | 0.44 | 0.39 | 0.33 | 0.05 | 0.25 | 1.56 | 1.07 | 0.02 | 4.53 | 2.09 | 1.69 |
|  | 2.95 | 2.75 |  | 0.03 | 6.4 | 1.21 | 1.08 | 0.72 | 0.81 | 0.41 | 0.05 | 5.52 | 0.15 | 0.34 | 0.63 | 0.38 | 0.32 | 0.05 | 0.14 | 1.56 | 0.34 | 0.05 | 0.55 | 0.92 | 0.92 |
|  | BULK |  |  |  | 105 | 6.7 | 15.7 | 11.39 | 7.90 | 0.96 | NA | 37.08 | 2.43 | 12.27 | 7.72 | 0.99 | 0.53 | 0.22 | 2.44 | 10.30 | 1.44 | 0.05 | 15.35 | 2.22 | 3.88 |
| N-2/ KWT-A2 | 1.40 | 1.60 | 1.60 | 0.02 | 419 | 11.5 | 36.3 | 25.28 | 6.81 | 1.19 | 0.15 | 97.92 | 3.34 | 3.06 | 1.55 | 0.33 | 0.08 | 0.07 | 0.50 | 2.06 | 0.71 | NA | 28.69 | 4.22 | 1.53 |
|  | 1.70 | 1.60 | 1.80 | 0.06 | 353 | 13.3 | 26.6 | 31.99 | 11.50 | 2.01 | 0.14 | 127.71 | 4.48 | 3.45 | 2.15 | 0.69 | 0.17 | 0.07 | 0.73 | 4.20 | 1.23 | NA | 42.30 | 5.66 | 2.60 |
|  | 1.90 | 1.80 | 2.00 | 0.16 | 198 | 12.1 | 16.4 | 30.44 | 18.46 | 2.80 | 0.31 | 112.80 | 5.97 | 3.42 | 2.59 | 1.03 | 0.28 | 0.10 | 1.02 | 8.81 | 1.73 | NA | 39.67 | 7.68 | 4.38 |
|  | 2.13 | 2.00 | 2.25 | 0.24 | 116 | 7.1 | 16.4 | 18.43 | 13.79 | 2.00 | 0.17 | 68.67 | 2.91 | 3.16 | 3.61 | 1.07 | 0.46 | 0.08 | 0.92 | 6.35 | 1.74 | NA | 33.96 | 6.14 | 4.09 |
|  | 2.38 | 2.25 | 2.50 | 0.21 | 19 | 1.2 | 15.2 | 3.26 | 3.60 | 0.82 | 0.10 | 18.70 | 0.60 | 1.43 | 1.40 | 1.64 | 0.25 | 0.02 | 0.37 | 2.69 | 0.98 | NA | 25.33 | 2.13 | 1.72 |
|  | 2.63 | 2.50 | 2.75 | 0.29 | 4.3 | 0.31 | 13.6 | 0.59 | 0.65 | 0.28 | 0.01 | 6.74 | 0.23 | 0.55 | 0.82 | 0.63 | 0.59 | 0.03 | 0.57 | 4.48 | 0.98 | NA | 6.01 | 2.13 | 2.81 |
|  | 2.95 | 2.75 |  | 0.03 | 6.0 | 0.30 | 13.1 | 0.96 | 0.51 | 0.34 | 0.04 | 2.60 | NA | 1.04 | 0.32 | 0.23 | 0.02 | 0.02 | 0.20 | 2.72 | 0.34 | NA | 0.85 | 0.39 | 0.97 |
|  | BULK |  |  |  | 105 | 5.6 | 20.1 | 10.44 | 7.90 | 0.36 | NA | 6.78 | NA | 3.57 | 1.89 | 0.71 | 0.33 | 0.03 | 1.38 | 7.52 | 0.35 | NA | 15.35 | 3.01 | 3.88 |
| N-3/ KWT-2A3 | 1.40 | 1.60 | 1.60 | 0.02 | 443 | 11.0 | 40.2 | 32.89 | 6.00 | 0.99 | 0.14 | 212.94 | 4.69 | 0.88 | 1.55 | 0.23 | 0.10 | 0.03 | 0.43 | 1.98 | 0.55 | NA | 35.89 | 3.49 | 1.42 |
|  | 1.70 | 1.60 | 1.80 | 0.07 | 353 | 10.5 | 33.7 | 48.76 | 8.89 | 2.21 | 0.15 | 228.72 | 6.89 | 4.77 | 1.64 | 0.79 | 0.25 | 0.04 | 0.64 | 4.09 | 1.23 | NA | 53.21 | 5.59 | 2.69 |
|  | 1.90 | 1.80 | 2.00 | 0.14 | 195 | 8.9 | 21.8 | 36.84 | 16.12 | 2.97 | 0.31 | 158.73 | 6.37 | 6.66 | 6.06 | 1.42 | 0.50 | 0.09 | 1.21 | 8.81 | 1.73 | NA | 44.89 | 8.23 | 5.60 |
|  | 2.13 | 2.00 | 2.25 | 0.25 | 112 | 6.4 | 17.6 | 26.20 | 16.20 | 2.63 | 0.32 | 102.53 | 4.60 | 6.66 | 3.61 | 1.64 | 0.65 | 0.08 | 1.19 | 9.64 | 2.05 | NA | 34.30 | 8.98 | 6.03 |
|  | 2.38 | 2.25 | 2.50 | 0.15 | 27 | 1.8 | 15.1 | 5.56 | 3.58 | 0.92 | 0.12 | 15.13 | 0.51 | 2.63 | 1.61 | 0.63 | 0.59 | 0.03 | 0.57 | 4.48 | 1.16 | NA | 25.33 | 3.44 | 2.81 |
|  | 2.63 | 2.50 | 2.75 | 0.37 | 4.5 | 0.30 | 14.1 | 1.02 | 0.65 | 0.30 | 0.04 | 2.24 | 0.23 | 0.55 | 0.82 | 0.12 | 0.23 | 0.02 | 0.20 | 2.72 | 0.34 | NA | 1.35 | 0.96 | 1.56 |
|  | 2.95 | 2.75 |  | 0.04 | 3.9 | 0.29 | 13.1 | 2.16 | 0.36 | 0.32 | 0.03 | 2.94 | 0.22 | 4.49 | 1.12 | 2.13 | 0.09 | 0.03 | 0.24 | 2.09 | 0.36 | NA | 2.91 | 7.05 | 3.30 |
|  | BULK |  |  |  | 98 | 4.9 | 22.2 | 8.39 | 8.39 | 0.78 | NA | 6.13 | NA | 11.31 | 4.33 | 2.95 | 0.31 | 0.08 | 1.85 | 6.26 | 0.47 | 0.02 | 24.38 | 7.07 | 4.83 |
| N-4/ KWT-3Bw | 1.40 | 1.60 | 1.60 | 0.02 | 181 | 7.8 | 23.1 | 8.99 | 6.48 | 1.49 | 0.06 | 91.14 | 4.35 | 56.80 | 13.73 | 23.49 | 0.61 | 0.43 | 2.07 | 1.34 | 1.08 | 0.04 | 27.21 | 63.66 | 2.74 |
|  | 1.70 | 1.60 | 1.80 | 0.07 | 289 | 12.9 | 22.4 | 7.80 | 6.48 | 1.19 | 0.06 | 62.68 | 3.18 | 62.68 | 16.06 | 31.39 | 0.74 | 0.98 | 1.72 | 4.09 | 1.02 | 0.04 | 17.25 | 70.71 | 5.46 |
|  | 1.90 | 1.80 | 2.00 | 0.22 | 71 | 8.1 | 23.9 | 9.88 | 7.80 | 1.19 | 0.07 | 39.38 | 1.73 | 31.39 | 16.06 | 31.39 | 0.63 | 0.83 | 1.21 | 7.50 | 2.72 | 0.05 | 42.41 | 62.64 | 6.11 |
|  | 2.13 | 2.00 | 2.25 | 0.13 | 119 | 4.9 | 24.6 | 7.81 | 9.57 | 0.82 | 0.07 | 29.80 | 1.33 | 15.71 | 26.68 | 15.71 | 0.55 | 1.69 | 0.39 | 8.83 | 2.93 | 0.06 | 36.31 | 5.86 | 2.33 |
|  | 2.38 | 2.25 | 2.50 | 0.12 | 39 | 3.3 | 11.7 | 2.70 | 1.38 | 0.48 | 0.01 | 3.80 | BD | 8.08 | 3.46 | 3.70 | 0.16 | 0.09 | 0.53 | 3.53 | 1.06 | NA | 22.21 | 4.81 | 2.30 |
|  | 2.63 | 2.50 | 2.75 | 0.48 | 9.0 | 0.75 | 12.2 | 0.98 | 0.36 | 0.34 | 0.01 | 0.96 | BD | 1.29 | 1.12 | 0.60 | 0.09 | 0.03 | 0.24 | 2.09 | 0.36 | NA | 1.71 | 1.85 | 2.65 |
|  | 2.95 | 2.75 |  | 0.04 | 1.9 | 0.15 | 12.2 | 2.16 | 1.50 | 0.32 | 0.03 | 2.94 | 0.22 | 4.49 | 26.59 | 2.13 | 0.40 | 0.08 | 0.17 | 6.26 | 0.47 | NA | 2.91 | 17.78 | 3.30 |
|  | BULK |  |  |  | 12 | 0.93 | 13.1 | 2.14 | 2.14 | 0.65 | 0.01 | 0.97 | 0.74 | 6.13 | 4.33 | 2.95 | 0.19 | NA | 0.30 | 0.82 | 0.47 | 0.02 | 6.56 | 13.47 | 4.83 |
| S-1/ MBB-CN | 1.40 | 1.60 | 1.60 | 0.04 | 418 | 16.2 | 25.9 | 30.29 | 30.29 | 0.73 | 0.06 | 121.29 | 3.66 | 0.52 | 2.38 | 0.08 | 0.08 | 0.10 | 0.05 | 0.85 | 1.08 | 0.47 | 30.29 | 1.71 | 0.47 |
|  | 1.70 | 1.60 | 1.80 | 0.17 | 289 | 12.9 | 22.4 | 62.17 | 6.48 | 0.52 | 0.06 | 131.37 | 3.18 | 0.74 | 4.66 | 0.12 | 0.12 | 0.08 | 0.21 | 2.24 | 0.68 | 1.34 | 25.9 | 3.07 | 1.34 |
|  | 1.90 | 1.80 | 2.00 | 0.22 | 193 | 8.1 | 23.9 | 65.06 | 7.80 | 1.19 | 0.07 | 104.88 | 2.40 | 0.98 | 6.45 | 0.19 | 0.20 | 0.05 | 0.28 | 2.95 | 0.53 | 1.76 | 42.41 | 4.20 | 1.76 |
|  | 2.13 | 2.00 | 2.25 | 0.13 | 119 | 4.9 | 24.6 | 36.31 | 9.57 | 0.27 | 0.07 | 74.52 | 1.34 | 1.42 | 8.87 | 0.30 | 0.30 | 0.04 | 0.39 | 3.86 | 0.68 | 2.33 | 36.31 | 5.86 | 2.33 |
|  | 2.38 | 2.25 | 2.50 | 0.04 | 61 | 2.6 | 23.7 | 4.58 | 1.38 | 0.48 | 0.01 | 40.66 | 0.65 | 6.91 | 8.87 | 0.52 | 0.16 | 0.02 | 0.40 | 4.50 | 0.78 | 2.65 | 22.21 | 4.81 | 2.65 |
|  | 2.50 | 2.50 |  | 0.27 | 5.8 | 0.35 | 16.5 | 0.35 | 0.36 | 0.14 | 0.01 | 1.71 | BD | 0.19 | 1.02 | 0.12 | 0.70 | 0.02 | 0.04 | 0.81 | 0.26 | 0.45 | 1.71 | 0.70 | 0.45 |
|  | 2.50 | 2.40 | 2.40 | 0.04 | 2.9 | 0.20 | 14.5 | 0.50 | 2.23 | 0.39 | 0.03 | 2.17 | BD | 0.82 | 5.65 | 0.84 | 0.09 | 0.03 | 0.24 | 2.86 | 0.70 | 1.59 | 2.91 | 3.85 | 3.30 |
|  | BULK |  |  |  | 146 | 5.8 | 25.0 | 6.94 | 35.91 | 1.31 | 0.01 | 68.78 | 1.34 | 11.31 | 4.76 | 2.95 | 0.19 | 0.03 | 0.30 | 6.26 | 0.47 | 0.02 | 24.90 | 3.55 | 1.42 |
| S-2/ MBB-DD | 1.40 | 1.60 | 1.60 | 0.02 | 359 | 11.5 | 31.2 | 33.23 | 9.39 | 1.43 | NA | 109.81 | 2.42 | 3.25 | 2.19 | 0.31 | 0.06 | 0.10 | 0.33 | 2.17 | 0.39 | NA | 37.92 | 4.35 | 1.41 |
|  | 1.70 | 1.60 | 1.80 | 0.02 | 278 | 11.4 | 24.3 | 43.76 | 23.34 | 3.94 | NA | 120.78 | 3.24 | 5.75 | 4.53 | 1.55 | 0.17 | 0.10 | 0.94 | 6.94 | 0.52 | NA | 55.43 | 8.02 | 4.41 |
|  | 1.90 | 1.80 | 2.00 | 0.06 | 179 | 8.3 | 21.6 | 33.84 | 26.64 | 2.77 | NA | 95.09 | 2.53 | 9.34 | 7.81 | 2.88 | 0.27 | 0.13 | 1.25 | 8.58 | 0.61 | NA | 47.16 | 13.25 | 5.54 |
|  | 2.10 | 2.00 | 2.20 | 0.11 | 103 | 5.0 | 20.5 | 33.84 | 65.06 | 1.79 | NA | 95.00 | 1.65 | 8.48 | 9.54 | 2.55 | 0.30 | 0.09 | 1.37 | 10.44 | 0.67 | NA | 58.85 | 13.25 | 6.58 |
|  | 2.30 | 2.20 | 2.40 | 0.16 | 53 | 2.4 | 22.0 | 12.17 | 13.74 | 1.27 | NA | 32.08 | 0.63 | 5.13 | 6.85 | 1.52 | 0.52 | 0.04 | 1.04 | 7.49 | 0.58 | NA | 19.04 | 8.56 | 6.58 |
|  | 2.50 | 2.40 | 2.60 | 0.43 | 11 | 0.54 | 19.6 | 2.61 | 3.59 | 0.40 | NA | 7.19 | BD | 0.82 | 1.43 | 0.24 | 0.82 | BD | 0.24 | 2.22 | 0.30 | NA | 4.41 | 1.53 | 4.79 |
|  | 2.80 | 2.60 |  | 0.12 | 8.0 | 0.45 | 17.8 | 0.91 | 1.33 | 0.24 | NA | 2.17 | BD | 0.52 | 1.47 | 0.24 | 1.25 | BD | 0.11 | 2.07 | 0.28 | NA | 1.58 | 1.25 | 1.15 |
|  | BULK |  |  |  | 59 | 2.4 | 24.1 | 11.25 | 10.82 | 1.21 | NA | 30.43 | 0.67 | 2.76 | 3.10 | 0.77 | 4.31 | 0.02 | 0.57 | 4.76 | 0.43 | 2.95 | 16.66 | 4.31 | 2.95 |

| Soil ID | Mean density (g cm⁻³) | Low cutoff (g cm⁻³) | High cutoff (g cm⁻³) | Mass fraction | OC (mg g⁻¹) | N (mg g⁻¹) | C/N | PP Al | PP Fe | PP Si | PP Mn | PP DOC | PP TDN | OX Al | OX Fe | OX Si | OX Mn | OX TDN | DC Al | DC Fe | DC Si | DC Mn | PP-extr | OX-extr Al+0.5Fe | DC-extr Fe |
|---|---|---|---|---|---|---|---|---|---|---|---|---|---|---|---|---|---|---|---|---|---|---|---|---|---|
| C-1/ NGK-Manure | 1.40 | | 1.60 | 0.01 | 238 | 12.9 | 18.4 | 1.15 | 1.86 | 2.34 | 0.12 | 17.05 | 1.28 | 1.93 | 1.48 | 0.67 | 0.02 | 0.09 | 1.01 | 3.42 | 2.51 | 0.01 | 2.08 | 2.67 | 2.72 |
| | 1.80 | 1.60 | 2.00 | 0.06 | 104 | 7.3 | 14.3 | 1.28 | 2.35 | 0.95 | 0.04 | 11.87 | 0.85 | 2.24 | 1.55 | 0.52 | 0.02 | 0.03 | 1.15 | 4.33 | 2.43 | 0.01 | 2.46 | 3.02 | 3.32 |
| | 2.25 | 2.00 | 2.50 | 0.40 | 28 | 2.8 | 10.1 | 1.32 | 2.56 | 0.48 | 0.02 | 6.26 | 0.56 | 3.47 | 2.54 | 0.62 | 0.03 | 0.03 | 2.08 | 8.73 | 2.34 | 0.01 | 2.60 | 4.74 | 6.45 |
| | 2.70 | 2.50 | | 0.51 | 1.7 | 0.17 | 10.1 | 0.23 | 0.32 | BD | BD | 0.87 | BD | 0.35 | 0.43 | 0.10 | 0.00 | BD | 0.46 | 2.52 | 0.33 | BD | 0.39 | 0.57 | 1.72 |
| | BULK | | | | 24 | 2.1 | 11.4 | 0.67 | 1.46 | 0.22 | 0.02 | 4.09 | 0.31 | 1.72 | 1.26 | 0.32 | 0.02 | 0.03 | 1.14 | 4.92 | 1.39 | 0.01 | 1.40 | 2.35 | 3.59 |
| C-2/ WKY-NPK | 1.40 | | 1.60 | 0.005 | 369 | 16.7 | 22.1 | 2.12 | 1.68 | 1.35 | 0.19 | 34.48 | 2.51 | 0.86 | 1.05 | 0.37 | 0.02 | 0.07 | 0.23 | 0.89 | 1.12 | 0.00 | 2.96 | 1.39 | 0.67 |
| | 1.70 | 1.60 | 1.80 | 0.003 | 356 | 17.2 | 20.7 | NA | NA | NA | NA | NA | NA | NA | NA | NA | NA | NA | NA | NA | NA | NA | NA | NA | NA |
| | 1.90 | 1.80 | 2.00 | 0.01 | 119 | 7.0 | 17.1 | 2.39 | 2.71 | 1.03 | 0.30 | 10.03 | 1.09 | 1.62 | 2.10 | 0.51 | 0.05 | 0.04 | 0.43 | 2.04 | 1.87 | 0.01 | 3.75 | 2.67 | 1.45 |
| | 2.13 | 2.00 | 2.25 | 0.02 | 39 | 3.3 | 11.9 | 2.73 | 3.90 | 0.70 | 0.43 | 3.67 | 0.91 | 2.11 | 2.24 | 0.74 | 0.21 | 0.03 | 0.42 | 1.91 | 2.31 | 0.02 | 4.67 | 3.24 | 1.37 |
| | 2.38 | 2.25 | 2.50 | 0.28 | 14 | 2.1 | 6.7 | 2.43 | 3.42 | 0.35 | 0.33 | 1.86 | 0.30 | 2.75 | 4.12 | 0.61 | 0.43 | BD | 0.55 | 2.12 | 1.58 | 0.04 | 4.14 | 4.81 | 1.61 |
| | 2.63 | 2.50 | 2.75 | 0.64 | 0.36 | 0.03 | 10.7 | 0.45 | 0.48 | 0.38 | 0.09 | BD | BD | 0.48 | 1.16 | 0.29 | 0.22 | BD | 0.25 | 1.71 | 0.45 | 0.02 | 0.69 | 1.06 | 1.10 |
| | 2.95 | 2.75 | | 0.02 | 0.29 | 0.03 | 10.7 | NA | NA | NA | NA | NA | NA | NA | NA | NA | NA | NA | NA | NA | NA | NA | NA | NA | NA |
| | BULK | | | | 9.8 | 1.0 | 9.4 | 1.29 | 1.68 | 0.45 | 0.30 | 1.96 | 0.30 | 1.39 | 2.17 | 0.46 | 0.17 | BD | 0.35 | 1.96 | 1.01 | 0.02 | 2.13 | 2.47 | 1.32 |
| C-3/ WKY-Straw | 1.40 | | 1.60 | 0.02 | 379 | 22.6 | 16.8 | 1.61 | 1.66 | 1.07 | 0.43 | 32.94 | 3.08 | 1.06 | 1.46 | 0.39 | 0.05 | 0.09 | 0.27 | 1.19 | 1.26 | 0.01 | 2.44 | 1.79 | 0.87 |
| | 1.70 | 1.60 | 1.80 | 0.02 | 274 | 18.5 | 14.8 | 1.92 | 2.71 | 1.10 | 0.95 | 20.67 | 1.95 | 1.89 | 2.46 | 0.80 | 0.13 | 0.07 | 0.44 | 1.65 | 1.57 | 0.03 | 3.28 | 3.12 | 1.27 |
| | 1.90 | 1.80 | 2.00 | 0.02 | 161 | 12.5 | 12.9 | 1.88 | 2.89 | 1.05 | 0.85 | 11.75 | 1.41 | 2.38 | 2.89 | 0.96 | 0.14 | 0.05 | 0.44 | 1.78 | 1.98 | 0.03 | 3.33 | 3.83 | 1.33 |
| | 2.13 | 2.00 | 2.25 | 0.03 | 61 | 5.3 | 11.5 | 2.58 | 5.25 | 0.72 | 0.62 | 6.96 | 0.94 | 2.87 | 3.96 | 0.98 | 0.15 | 0.05 | 0.50 | 1.90 | 2.36 | 0.04 | 5.20 | 4.85 | 1.45 |
| | 2.38 | 2.25 | 2.50 | 0.25 | 22 | 2.8 | 7.9 | 1.40 | 2.42 | 0.51 | 0.41 | 2.32 | 0.18 | 2.81 | 3.84 | 0.83 | 0.30 | 0.02 | 0.53 | 1.85 | 1.76 | 0.06 | 2.61 | 4.73 | 1.46 |
| | 2.63 | 2.50 | 2.75 | 0.60 | 0.66 | 0.06 | 10.3 | 0.30 | 0.33 | 0.27 | 0.10 | 0.00 | BD | 0.41 | 1.21 | 0.28 | 0.23 | 0.01 | 0.23 | 1.80 | 0.45 | 0.03 | 0.47 | 1.02 | 1.13 |
| | 2.95 | 2.75 | | 0.02 | 0.53 | 0.05 | 10.3 | NA | NA | NA | NA | NA | NA | NA | NA | NA | NA | NA | NA | NA | NA | NA | NA | NA | NA |
| | BULK | | | | 29 | 2.5 | 11.7 | 1.06 | 1.97 | 0.61 | 0.44 | 2.35 | 0.53 | 1.41 | 2.77 | 0.59 | 0.12 | 0.02 | 0.39 | 2.47 | 1.14 | 0.04 | 2.04 | 2.80 | 1.63 |
| C-4/ MNG-A | 1.40 | | 1.60 | 0.02 | 250 | 17.5 | 14.3 | 1.52 | 0.50 | 0.96 | 0.18 | 20.10 | 1.31 | 1.49 | 1.92 | 0.28 | 0.13 | 0.10 | 1.01 | 6.21 | 2.68 | 0.04 | 1.78 | 2.44 | 4.11 |
| | 1.70 | 1.60 | 1.80 | 0.01 | 261 | 20.5 | 12.7 | 2.06 | 0.92 | 1.35 | 0.25 | 30.00 | 1.89 | 2.06 | 2.77 | 0.68 | 0.22 | 0.05 | 0.32 | 1.37 | 1.84 | 0.02 | 2.52 | 3.45 | 1.01 |
| | 1.90 | 1.80 | 2.00 | 0.03 | 172 | 16.6 | 10.3 | 1.80 | 0.79 | 0.82 | 0.25 | 21.48 | 1.37 | 2.45 | 3.23 | 0.48 | 0.30 | 0.05 | 1.60 | 10.97 | 3.95 | 0.07 | 2.19 | 4.07 | 7.08 |
| | 2.10 | 2.00 | 2.20 | 0.04 | 92 | 11.0 | 8.3 | 1.96 | 0.91 | 0.80 | 0.16 | 14.96 | 0.99 | 3.11 | 4.02 | 0.62 | 0.34 | 0.08 | 1.49 | 13.96 | 3.64 | 0.04 | 2.41 | 5.12 | 8.47 |
| | 2.30 | 2.20 | 2.40 | 0.09 | 49 | 6.6 | 7.5 | 1.44 | 0.83 | 0.75 | 0.14 | 8.83 | 0.49 | 3.54 | 4.69 | 0.87 | 0.43 | 0.07 | 1.67 | 16.58 | 3.98 | 0.07 | 1.85 | 5.88 | 9.96 |
| | 2.50 | 2.40 | 2.60 | 0.74 | 1.7 | 0.34 | 4.9 | 0.22 | 0.09 | 0.21 | 0.02 | 0.39 | BD | 0.31 | 0.40 | 0.15 | 0.04 | 0.01 | 0.23 | 1.27 | 0.45 | 0.13 | 0.27 | 0.51 | 0.87 |
| | 2.80 | 2.60 | | 0.07 | 2.8 | 0.89 | 3.1 | 0.49 | 0.27 | 0.19 | 0.03 | 0.66 | BD | 0.91 | 2.88 | 0.46 | 0.10 | 0.03 | 0.38 | 6.67 | 0.87 | 0.01 | 0.62 | 2.35 | 3.72 |
| | BULK | | | | 24 | 2.5 | 9.7 | 0.49 | 0.25 | 0.44 | NA | NA | NA | 1.09 | 1.22 | 0.31 | 0.12 | NA | 0.56 | 4.52 | 1.30 | NA | 0.62 | 1.70 | 2.82 |
| C-5/ KIN-07S | 1.40 | | 1.60 | 0.01 | 315 | 11.0 | 28.5 | 4.93 | 5.46 | 1.17 | 0.02 | 39.47 | 2.14 | 1.26 | 2.65 | 0.37 | BD | 0.07 | 0.25 | 1.09 | 0.79 | 0.00 | 7.66 | 2.59 | 0.79 |
| | 1.80 | 1.60 | 2.00 | 0.02 | 162 | 7.7 | 20.9 | 3.46 | 10.96 | 0.67 | 0.01 | 27.58 | 1.01 | 0.99 | 2.20 | 0.17 | BD | 0.05 | 2.40 | 17.00 | 1.18 | 0.01 | 8.94 | 2.09 | 10.90 |
| | 2.10 | 2.00 | 2.20 | 0.03 | 51 | 4.1 | 12.5 | 1.97 | 8.42 | 0.53 | 0.01 | 12.53 | 0.70 | 1.12 | 2.54 | 0.19 | BD | 0.03 | 2.28 | 15.78 | 1.54 | 0.01 | 6.18 | 2.39 | 10.17 |
| | 2.30 | 2.20 | 2.40 | 0.16 | 37 | 3.6 | 10.3 | 2.29 | 10.43 | 0.21 | 0.01 | 12.40 | 0.57 | 1.78 | 4.87 | 0.19 | BD | 0.05 | 3.49 | 23.48 | 1.03 | 0.01 | 7.50 | 4.22 | 15.23 |
| | 2.50 | 2.40 | 2.60 | 0.31 | 12 | 1.1 | 10.8 | 1.00 | 5.92 | 0.14 | BD | 5.85 | 0.38 | 0.53 | 2.05 | 0.06 | BD | 0.02 | 1.24 | 8.48 | 0.41 | BD | 3.95 | 1.56 | 5.48 |
| | 2.80 | 2.60 | | 0.47 | 2.6 | 0.32 | 8.3 | 0.27 | 1.56 | 0.14 | BD | 1.99 | BD | 0.15 | 0.73 | 0.02 | BD | BD | 0.41 | 3.48 | 0.12 | BD | 1.05 | 0.51 | 2.15 |
| | BULK | | | | 21 | 1.9 | 11.1 | 1.14 | 3.64 | 0.12 | BD | 1.70 | 0.58 | 0.79 | 2.13 | 0.08 | BD | 0.06 | 1.53 | 10.75 | 0.51 | NA | 2.96 | 1.86 | 6.91 |
| C-6/ KIN-17S | 1.40 | | 1.60 | 0.02 | 353 | 13.1 | 26.9 | 3.78 | 6.57 | 0.52 | 0.01 | 52.69 | 2.32 | 0.69 | 1.82 | 0.08 | BD | 0.18 | 1.10 | 7.65 | 0.62 | BD | 7.06 | 1.60 | 4.93 |
| | 1.70 | 1.60 | 1.80 | 0.03 | 265 | 13.2 | 20.1 | 3.43 | 7.15 | 0.75 | 0.01 | 65.63 | 2.86 | 0.47 | 1.35 | 0.09 | BD | 0.13 | 1.19 | 8.59 | 0.91 | BD | 7.00 | 1.15 | 5.49 |
| | 1.90 | 1.80 | 2.00 | 0.03 | 214 | 11.4 | 18.8 | 2.92 | 8.81 | 0.75 | 0.01 | 53.72 | 2.73 | 0.54 | 1.64 | 0.11 | BD | 0.15 | 1.60 | 11.85 | 1.10 | BD | 7.33 | 1.36 | 7.53 |
| | 2.10 | 2.00 | 2.20 | 0.11 | 144 | 8.9 | 16.2 | 2.27 | 11.00 | 0.62 | BD | 36.69 | 1.96 | 0.72 | 2.17 | 0.15 | BD | 0.11 | 1.93 | 14.49 | 1.40 | BD | 7.77 | 1.80 | 9.18 |
| | 2.30 | 2.20 | 2.40 | 0.08 | 81 | 5.0 | 16.2 | 1.61 | 8.55 | 0.42 | BD | 23.24 | 1.18 | 0.58 | 2.14 | 0.09 | BD | 0.07 | 1.46 | 10.89 | 0.95 | BD | 5.88 | 1.65 | 6.90 |
| | 2.50 | 2.40 | 2.60 | 0.59 | 4.0 | 0.30 | 13.3 | 0.09 | 0.70 | 0.11 | BD | 1.91 | BD | 0.01 | 0.22 | 0.01 | BD | BD | 0.07 | 0.94 | 0.07 | BD | 0.44 | 0.12 | 0.54 |
| | 2.75 | 2.60 | | 0.14 | 6.0 | 0.30 | 20.0 | 0.13 | 0.82 | 0.10 | BD | 1.95 | BD | 0.07 | 0.65 | 0.02 | BD | BD | 0.75 | 9.21 | 0.12 | BD | 0.54 | 0.39 | 5.35 |
| | BULK | | | | 42 | 2.5 | 16.8 | 1.17 | 3.65 | 0.12 | 0.01 | 1.73 | 0.58 | 0.27 | 0.86 | 0.02 | BD | 0.02 | 1.40 | 7.30 | 0.70 | BD | 3.00 | 0.70 | 5.05 |
| C-7/ KIN-27S | 1.40 | | 1.60 | 0.15 | 388 | 16.6 | 23.4 | 1.69 | 3.60 | 1.43 | 0.03 | 39.15 | 2.00 | 0.31 | 1.06 | 0.13 | BD | 0.12 | 0.19 | 1.65 | 0.59 | BD | 3.49 | 0.84 | 1.01 |
| | 1.70 | 1.60 | 1.80 | 0.11 | 271 | 17.9 | 15.2 | 1.52 | 3.43 | 1.97 | 0.01 | 47.73 | 2.44 | 0.55 | 0.61 | 0.23 | BD | 0.10 | 0.28 | 2.33 | 1.70 | BD | 3.24 | 0.86 | 1.44 |
| | 1.90 | 1.80 | 2.00 | 0.10 | 174 | 12.1 | 14.4 | 0.75 | 1.86 | 1.79 | BD | 25.71 | 1.33 | 0.62 | 0.49 | 0.24 | BD | 0.07 | 0.30 | 2.49 | 2.01 | BD | 1.68 | 0.86 | 1.55 |
| | 2.10 | 2.00 | 2.20 | 0.04 | 115 | 8.8 | 13.1 | 0.63 | 1.41 | 1.52 | BD | 15.07 | 0.82 | 0.76 | 0.62 | 0.26 | BD | 0.06 | 0.40 | 3.12 | 1.94 | BD | 1.33 | 1.07 | 1.96 |
| | 2.30 | 2.20 | 2.40 | 0.03 | 76 | 6.0 | 12.7 | 0.54 | 1.89 | 0.59 | 0.01 | 9.69 | 0.52 | 0.90 | 1.07 | 0.18 | BD | 0.06 | 0.53 | 3.60 | 0.90 | BD | 1.48 | 1.44 | 2.33 |
| | 2.50 | 2.40 | 2.60 | 0.05 | 12 | 1.3 | 9.2 | 0.12 | 0.74 | 0.16 | BD | 0.98 | BD | 0.41 | 0.94 | 0.08 | 0.01 | 0.02 | 0.36 | 3.08 | 0.32 | BD | 0.49 | 0.88 | 1.90 |
| | 2.80 | 2.60 | | 0.49 | 1.5 | 0.15 | 9.8 | 0.03 | 0.15 | 0.07 | BD | 0.00 | BD | 0.03 | 0.21 | 0.01 | 0.01 | BD | 0.04 | 1.19 | 0.04 | BD | 0.11 | 0.14 | 0.63 |
| | BULK | | | | 125 | 8.4 | 14.9 | 0.68 | 1.54 | 1.15 | 0.07 | 19.22 | 1.48 | 0.25 | 0.46 | 0.09 | NA | NA | 0.13 | 1.60 | 0.85 | NA | 1.45 | 0.48 | 0.93 |

NA: not available. BD: below detection. * The metal extraction was not conducted for the lowest-density fraction of A-8 soil due to the very small recovery of this fraction. We thus assumed that the extractable element concentrations in this fraction were the same as those in the A-9 soil, which is adjacent to the A-8 plot as a part of the long-term field experiment in Yamanashi, Japan.

**Table A2.** The results of a cubic polynomial CE2 regression model applied to each of the four soil groups for pyrophosphate-extractable organic C ($DOC_{PP}$) and the sum of pyrophosphate- and oxalate-extractable C ($DOC_{PP} + DOC_{OX}$).

| | Allophanic Andisol | Non-allophanic Andisol | Spodic | Crystalline mineralogy |
|---|---|---|---|---|
| $DOC_{PP}$ | | | | |
| $r^2$ | 0.17 | 0.41 | 0.55 | 0.50 |
| RMSE | 0.12 | 0.24 | 0.12 | 0.11 |
| $n$ | 49 | 27 | 14 | 39 |
| $F$ | 3.13 | 5.21 | 4.05 | 11.86 |
| $p$ | 0.0347 | 0.0068 | 0.040 | < 0.0001 |
| $DOC_{PP} + DOC_{OX}$ | | | | |
| $r^2$ | 0.32 | 0.47 | 0.55 | 0.51 |
| RMSE | 0.12 | 0.22 | 0.12 | 0.11 |
| $n$ | 49 | 27 | 14 | 39 |
| $F$ | 6.99 | 6.73 | 4.12 | 12.27 |
| $p$ | 0.0006 | 0.0020 | 0.0384 | < 0.0001 |

https://doi.org/10.5194/soil-6-1-2020

**Data availability.** .TS14

**Author contributions.** RW designed the experiments that were carried out by all the co-authors. KM developed the mathematical approach to assess the density-dependent metal distribution. RW prepared the manuscript with contributions from all the co-authors.

**Competing interests.** The authors declare that they have no conflict of interest.TS15 .

**Acknowledgements.** This study was supported by JSPS KAKENHI grant nos. 15KK0028, 15H02810, 15KT0036, and 18K05369 and MAFF-JPJ008837, JapanTS16 . We gratefully thank R. MatsuuraTS17 (Tokyo Metropolitan Agriculture and Forestry Research Center), K. Maruoka (Saitama Agricultural Technology Research Center), S. Yamazaki (Yamanashi Prefectural Agricultural Technology Center), H. Wakasawa (Shizuoka Prefectural Research Institute of Agriculture and Forestry), M. Kasuya (Aichi Agricultural Research Center), and Y. Hayashi (Wakayama Research Center of Agriculture, Forestry and Fisheries) for kindly sharing soils from their long-term experimental fields. We also thank I. J. Fernandez and T. Ohno (Univ. of Maine), S. Hiradate (Univ. of Kyushu), T. Tanikawa (Univ. of Nagoya), K. Kitayama (Univ. of Kyoto), and the Sabah Park for field support and Y. Yaegaki and C. Hayakawa for laboratory assistance. Comments by L. M. Mayer on an earlier draft are greatly thanked.

**Financial support.** This research has been supported by the JSPS (Japan Society for the Promotion of Science) (grant nos. 15KK0028, 15H02810, 15KT0036, 18K05369, and MAFF-JPJ008837, Japan).TS18

**Review statement.** This paper was edited by Cornelia Rumpel and reviewed by two anonymous referees.

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

**Remarks from the language copy-editor**

CE1    Please confirm.

CE2    Please confirm.

**Remarks from the typesetter**

TS1    Please provide a suitable running title.

TS2    Please add a reference to the bibliography.

TS3    This citation has no corresponding reference list entry.

TS4    Is the colon here a symbol for division or for a ratio?

TS5    This citation has no corresponding reference list entry.

TS6    This citation has no corresponding reference list entry.

TS7    This citation has no corresponding reference list entry.

TS8    This citation has no corresponding reference list entry.

TS9    This citation has no corresponding reference list entry.

TS10    This citation has no corresponding reference list entry.

TS11    The composition of Fig. A3 has been adjusted to our standards.

TS12    Please check the layout of this table carefully.

TS13    One instance of "fraction" has been deleted.

TS14    Please provide a statement on how your underlying research data can be accessed. If the data are not publicly accessible, a detailed explanation of why this is the case is required. The best way to provide access to data is by depositing them (as well as related metadata) in reliable public data repositories, assigning digital object identifiers (DOIs), and properly citing data sets as individual contributions. Please indicate if different data sets are deposited in different repositories or if data from a third party were used. Additionally, please provide a reference list entry including creators, title, and date of last access. If no DOI is available, assets can be linked through persistent URLs to the data set itself (not to the repositories' home page). This is not seen as best practice and the persistence of the URL must be secured.

TS15    Declaration of all potential conflicts of interest is required by us as this is an integral aspect of a transparent record of scientific work. If there are possible conflicts of interest, please state what competing interests are relevant to your work. Please see https://publications.copernicus.org/services/competing_interests_policy.html

TS16    Please confirm changes.

TS17    Please provide full first names throughout this section.

TS18    Please note that the funding information has been added to this paper. Please check if it is correct. Please also double-check your acknowledgements to see whether repeated information can be removed or changed accordingly. Thanks.

TS19    Please provide page range or DOI and article number.

TS20    Please provide place of publication.

TS21    Please provide full editor names.

TS22    Please provide place of publication.

TS23    Please provide place of publication.

TS24    Please provide place of publication.

TS25    Please provide page range or article number.

TS26    Please provide page range or DOI and article number.

TS27    Please provide place of publication.

TS28    Please check DOI and provide volume and page range or article number.

TS29    Please provide publisher and place of publication.

TS30    Please provide page range or DOI and article number.