# Peer review of "Iron and aluminum association with microbially processed organic matter via meso-density aggregate formation across soils: organometallic glue hypothesis"

_SOIL, 2020_

## Referee Comment (RC1) · Anonymous Referee #1 · 6 Jul 2020

The paper "Iron and aluminum association with microbially processed organic matter via meso-density aggregate formation across soils: organometallic glue hypothesis" proposed by Rota Wagai and co-authors shows the fundamental role that organometallic complexes play for two essential soil functions:

- interactions with OM that contribute to soil C stock and

- their structuring role in aggregation processes.

The soils used allow the hypotheses to be tested on 4 different soils. They allow to test

different mineralogies and different climates. This work is based on an experimental approach by densimetric fractionation. The number of fractionated samples is very important. The experimental component therefore represents a large-scale laboratory work that seems to have been carried out rigorously.

The results are presented in a very synthetic and clear manner. The article is well written and well structured. The discussion is well conducted and convincing.

I really appreciated this work and I congratulate the authors. It's a fairly original approach that seeks to bring together knowledge on the mechanisms of organo-mineral interactions with aggregation processes. In this sense, it seems to me that it is an innovative approach that certainly deserves to be published without delay. Moreover, this article is one in a series of recent innovative papers that show the importance of poorly crystallized mineral phases and metals in soil C dynamics.

However, there are a few points that I think can improve the paper. My main comments to be taken into account in a revised version are:

- It is essential that the authors provide all data for each fraction of each soil. It is impossible to verify the calculations proposed by the authors, it is impossible to know what are the losses during densimetric fractionation for individual soils, it is impossible to re-use the data for other works. My recommendation is to provide a summary table of all the data as additional information.

- The title needs to be changed because evidence that the organic matter of "nanocomposites" has been biotransformed by the microbial compartment is "light". This argument is only based on the value of C:N which shows quite variable values (shown only in the supplementary informations graphs). Even if the trend is probably true, the authors have not investigated enough to state it with certainty.

- In the introduction, last paragraph: the scientific questions asked must be made explicit.

In details :

Abstract :

- Indicate the regions of origin of the samples.

- L 24 -27 Sentence too long

- L27 : remove stable : there's no argument that the OM is "stable."

Methods

- The soils choices could be better justified.

- Give more details on the fractionation protocol.

- Explain why the number of fractions is different from one soil to another?

- SPT is very acidic (pH down to 2); what could be the effect of such a pH on the "nanocomposites" ?

Chemical analyses

- Instead of adding weight bas concentration (Al+1/2 Fe) to approximately normalize the atomic mass difference between Al and Fe, working with atomic concentration would be more rigorous!

Peak density determination

- To my mind, "peak density" is a term which is not really appropriate

Result

- I would suggest to first present the data (see previous remark) before presenting the recovery

- L230 Fig A1 : left panel

- L290 : I may add an additional sentence to be sure that the reader understand properly the difference between fig 2 concentration and fig 5 distribution

Discussion - L326 what are the "non-centrifugeable colloidal Fe/ Al oxide phases?

- L328-331 : no data on Si were provided in the result part. (I think you may add a short section also on Si in the results)

- L345 : Is a graph Al:Fe as a function of pH interesting? I would have enjoyed to see it !

- "nanocomposite" could also have been called "nanoCLICS" as proposed by Tamrat et al. 2019.

- Fig 7a : position of the peaks is not consistent with fig 5

- I do enjoy Fig 7 b !

---

## Referee Comment (RC2) · Anonymous Referee #2 · 22 Jul 2020

Wagai and co-workers present a thorough, comprehensive study in which they couple a multi-stage density fractionation on 23 different soils with selective Fe-fractionations. The results support their hypothesis that most of the organic material is contained in aggregates of Al/Fe complexes, Al/Fe oxides and clay minerals. The validity of the study is of course strongly dependent on the selectivity and other artifacts of the fractionation methods. However, the authors give high recovery rates and discuss possible errors in great detail. I support publication with very minor revisions.

Abstract: I think that the term "density fraction locations" (Line 12) and the sentence in

lines 17-18 are hard to understand before the article is read. Please consider alternative phrases.

Line 21: Please include that the assumption of having microbial processed OM is based on C/N ratios

Line 27: why is the OM supposed to be stabile?

Manuscript: Line 141: Why were the fractions dried using different methods?

Line 291; Figure 6: I wonder why all SEM images look so similar. Everything seems to be aggregated. Is that also the case for the original soils? Or is it possible that the polytungstate treatment promotes aggregation?

Figure caption: where do you see clay coatings? Can you add arrows?

Line 428: delete "extractable"?

Line 506: correct "understanding"

---

## Referee Comment (RC3) · Anonymous Referee #2 · 23 Jul 2020

With respect to the abstract: I suggest to write that the material is "microbially processed" (important observation!), but to mention that this is assumed due to the C/N ratios. (In the discussion the point is well explained, but I missed it in the abstract.)
* * *

---

## Short Comment (SC1) · 23 Jul 2020

Thank you for the through review and valuable comments.

First of all, I appreciate your overall comment on our work ("to bring together knowledge on the mechanisms of organo-mineral interactions with aggregation processes"). That was exactly what we attempted to do.

Second, all of your specific comments are well-taken. We think we can address all the points.

[Figure]

One question raised was our interpretation of "microbially-processed OM" in the title and other parts on the ground of relatively low C:N ratio alone. We agree that we have to cautious on this and debated on how to describe this. We think the OM in the meso-density fractions are the mixture of both plant- and microbially-derived OM as depicted in Fig. 7b (glad that you liked this!). That was why we did not call it "microbially-derived". We have N-15 results from these samples that showed the enrichment in the meso-density relative to bulk and low-density fractions for each soil (this data is more complex and the current manuscript was already too long. we work on a new manuscript for this). So, both C:N and N-15 suggest strong "microbial influence". Do you think it is more appropriate to remove "microbially-processed"?

Al+0.5Fe: This expression is also something we debated. I agree with the reviewer that atomic (molar) expression is more scientific/general. This would make it easier to compare results across different systems (e.g., aquatic system). On the other hand, OC is often expressed on wt basis and readers are not so familiar with C content on atomic mass basis. So when we compare OC and metal, we have to choose. In main text, we included molar ratio information (e.g., Al/Fe). But maybe we should change expression more. Let us think a bit more.

We should be able to address other points easily. We will prepare an official response letter which explains how we address each point.

Thank you again for the through review and constructive comments.

Rota Wagai

---

## Short Comment (SC2) · 23 Jul 2020

Thank you for the through review and careful comments.

The expression of "microbially-processed OM" based solely on C:N ratio was also questioned by Referee#1. As I posted the reply comment to Ref#1, we are debating if we should remove "microbially-processed" or not. Please take a look at our response if possible. In brief, our separate study (in prep.) showed N-15 enrichment in meso-density OM relative to bulk and low-density fractions.

SEM images: We were also initially surprised by the well-aggregated appearance of all these fractions. But the aggregate nature observed by SEM is fairly consistent with the results of other experiments (unpublished) that show that most materials in the meso-density fractions are dispersible only after much stronger energy (by sonication). That was done right after the density fractionation without drying fractionated samples. So we don't think the aggregated nature is an artefact. Clay coatings are the plate-like features you see on most of these aggregates. We will try preparing higher-resolution images for the revision.

We should be able to address other points as well. We will prepare an official response letter. Also thank you for pointing out the typo and unclear parts.

Rota Wagai
* * *

---

## Short Comment (SC3) · 23 Jul 2020

Thank you for the clarification very quickly.

OK, we plan to keep "microbially-processed" in the title provided that Referee #1 agrees. In any case, we will update Abstract to clarify why think the material is microbially-processed by adding C:N ratio information.

---

## Author Comment (AC1) · 28 Jul 2020

Thank you for the thorough review, encouragement, and valuable constructive comments. We responded to each of your comments below.

Rota Wagai (on behalf of the authors)

– It is impossible to verify the calculations proposed by the authors, it is impossible to know what are the losses during densimetric fractionation for individual soils, it is impossible to re-use the data for other works. My recommendation is to provide a

summary table of all the data as additional information.

—> We did not provide individual data as we tried to limit the volume of the study. In revision, we can provide the values for individual soil density fractions as a table (Table A1).

– The title needs to be changed because evidence that the organic matter of "nanocomposites" has been biotransformed by the microbial compartment is "light". This argument is only based on the value of C:N which shows quite variable values (shown only in the supplementary informations graphs). Even if the trend is probably true, the authors have not investigated enough to state it with certainty.

—> We agree that we have to be careful on this point. We think the OM in the meso-density fractions is the mixture of both plant- and microbially-derived OM as depicted in Fig. 7b (glad that you liked this!). That was why we did not call it "microbially-derived".

However, we think it is appropriate to call it "microbially-processed" for three reasons: (i) the C:N ratio of the meso-density fractions was consistently lower by 2-23 units relative to the lowest-density fraction which mainly consists of plant detritus. (ii) SEM observation showed decreasing abundance of plant detritus with increasing density, (iii) delta N-15 analysis also showed that the meso-density OM was always more enriched relative to the OM in the low-density fraction by ca 2-6 per mil (unpublished data). The N-15 analysis of the density fractions was done for 14 out of the 23 soil samples. This result is more complex to interpret and will be incorporated into our next work. For these reasons, we think it is more appropriate to interpret that the majority of the OM present in the meso-density fractions is "microbially-processed".

We will revise the abstract section related to this topic as follows (L21-24). "The OM in meso-density fractions showed 2-23 unit lower C:N ratio than the lowest-density fraction of respective soil and thus appeared microbially processed from the original plant material."

- In the introduction, last paragraph: the scientific questions asked must be made explicit.

—> We think the questions asked are clear enough (one hypothesis and several related questions). Or do you mean the last paragraph where we define the terminology?

- Abstract : - Indicate the regions of origin of the samples.

—> the information will be added in Abstract as follows: "We identified density fraction locations of major metal phases and OM using 23 soil samples from 5 climate zones and 5 soil orders (Andisols, Spodosols, Inceptisols, Mollisols, Ultisols) from Asia and North America, including . . ...".

- L 24 -27 Sentence too long

—> We agree that it is a long sentence. We thought hard but we cannot think of a good way to shorten it at this moment.

- L27 : remove stable : there's no argument that the OM is "stable."

—> "OM" is now removed.

Methods - The soils choices could be better justified.

—> We now explained the rational for the soil samples we selected in L134-136.

- Give more details on the fractionation protocol.

—> Now the protocol is fully described.

- Explain why the number of fractions is different from one soil to another?

—> Now it is explained.

- SPT is very acidic (pH down to 2); what could be the effect of such a pH on the "nanocomposites" ?

—> We will add the following sentence at the end of density fractionation section under

Method. "We also assume little impact of sodium polytungstate on the extractability of Fe and Al phases or the nature of soil microaggregates as the SPT solution after the density fractionation typically had the pH value similar to bulk soil pH."

Chemical analyses - Instead of adding weight bas concentration (Al+1/2 Fe) to approximately normalize the atomic mass difference between Al and Fe, working with atomic concentration would be more rigorous!

—> We debated on this point. We agree that the expression in the atomic mass is more rigorous and facilitate the comparison with other materials (e.g., experimental mixture, sediments). But the major interest is the comparison between these metals and C. And the C concentration in soils and soil physical fractions is almost always expressed on a weight basis. So we would like to keep our unit as is. But we now also provide these values on atomic mass in appropriate sections (figure legend in Fig. 1 and 3, the main text where we explain OC:metal ratios) to allow such comparison.

So, in the revised manuscript, we plan to add a following sentences in Method section (L189). "This allows us to compare the metal values with C on a weight basis. We also reported some values including the stoichiometric relationships among the target elements (e.g., Al:Si ratio) on molar basis."

Peak density determination - To my mind, "peak density" is a term which is not really appropriate

—> We cannot think of any better ways to explain this. Sorry. . . We will add more explanation the corresponding parts in the main text to enhance the clarity to on this expression.

Result - I would suggest to first present the data (see previous remark) before presenting the recovery

—> We think it is better to show the recovery data first. If the recovery is now good, then the quality of the rest of results becomes questionable. So we prefer to put this

information upfront as previous studies did (e.g., Swanston et al., 2005, Geoderma) https://www.sciencedirect.com/science/article/pii/S0016706104003258

- L230 Fig A1 : left panel - L290 : I may add an additional sentence to be sure that the reader understand properly the difference between fig 2 concentration and fig 5 distribution

—> Thank you for the advice. We will add some words to improve the clarity of this first sentence.

Discussion - L326 what are the "non-centrifugeable colloidal Fe/ Al oxide phases?

—> This refers to the colloids that were too small to spin down by the centrifugation used. This corresponds to the limitation of PP extraction discussed 11 lines above this sentence in the same paragraph.

- L328-331 : no data on Si were provided in the result part. (I think you may add a short section also on Si in the results)

—> Thank you for pointing out. We now will report Si results in Result section by reporting (1) Al:Si molar ratios of each density fraction for each soil, and (2) the proportion of total extractable Si in pyrophosphate- and oxalate-extractable phases.

- L345 : Is a graph Al:Fe as a function of pH interesting? I would have enjoyed to see it !

—> I agree that it would be interesting! But we don't have pH of each density fraction. So we cannot do this.

- "nanocomposite" could also have been called "nanoCLICS" as proposed by Tamrat et al. 2019.

—> We will add this acronym where we cited this work in Discussion section.

- Fig 7a : position of the peaks is not consistent with fig 5

—> The Y axis in Figure 7a is the concentration of metal per fraction where as Fig. 5 showed the metal distribution. So Fig. 7a matches with Fig. 2.

We realized that it is easy for readers to get confused with two expressions (concentration and distribution). So we plan to add some sentences to clearly distinguish the two expressions in Method as well as Results/Discussion sections.

- I do enjoy Fig 7 b ! —> Very glad that you liked it!

---

## Author Comment (AC2) · 28 Jul 2020

Anonymous Referee #2 Thank you for the thorough review and positive comments. Below we show how we plan to address the points raised.

Rota Wagai, on behalf of the authors

Abstract: I think that the term "density fraction locations" (Line 12) and the sentence in lines 17-18 are hard to understand before the article is read. Please consider alternative phrases.

[Figure]

*** We agree that "density fraction locations" is not a familiar term but we think it is reasonable because it is logically the same as "size fraction locations". *** We will modify the sentence in line 17-18 to make it easier to understand for readers. The new sentence proposed is as follows.

"The concentrations of Fe and Al (per fraction) extracted by each of the three reagents able Fe and Al concentrations were (per fraction) generally higher in meso-density fractions (1.8-2.4 g cm-3) than in the lower- or higher-density fractions, showinged unique unimodal pattern distribution along particle density gradient for each soil and each extractable metal phase. "

Line 21: Please include that the assumption of having microbial processed OM is based on C/N ratios

*** Yes, we will add the information on C:N ratio.

Line 27: why is the OM supposed to be stabile?

*** We agree that it was arbitrary. We will remove the word "stable".

Manuscript: Line 141: Why were the fractions dried using different methods?

*** We plan to explain the reason and the assumption behind as follows in Method section.

"The lowest-density fractions (<1.6 g cm-3) were oven-dried at 80 oC instead of freeze-drying for a logistical reason. Due to the concentration of the extractable metals in this fraction, we assumed little effect of the difference in the drying method on our result interpretation.

Line 291; Figure 6: I wonder why all SEM images look so similar. Everything seems to be aggregated. Is that also the case for the original soils? Or is it possible that the polytungstate treatment promotes aggregation? Figure caption: where do you see clay coatings? Can you add arrows?

\*\*\* We were also somewhat surprised by the well-aggregated appearance of all these fractions. But the aggregate nature observed by SEM is fairly consistent with the results of other experiments (unpublished) that show that most materials in the meso-density fractions are dispersible only after much stronger energy (by sonication). That was done right after the density fractionation without drying fractionated samples. So we don't think the aggregated nature is an artifact. Clay coatings are the plate-like features you see on most of these aggregates. We will try preparing higher-resolution images for the revision.

Line 428: delete "extractable"?

\*\*\* Thank you for pointing out. It will be deleted.

Line 506: correct "understanding"

\*\*\*We will correct.

---

## Author Response (AR1)

Dear Editor,

Our responses to each of the comments from the two referees are shown below. We also include the revised manuscript with the tracks of changes we made. In addition to the responses to the referee's comments, we also made some minor edits to correct mistakes and enhance readability.

Rota Wagai (on behalf of the authors)

**Anonymous Referee #1**

Thank you for the through review, encouragement, and valuable constructive comments. We responded to each of your comments below.

*-- It is impossible to verify the calculations proposed by the authors, it is impossible to know what are the losses during densimetric fractionation for individual soils, it is impossible to re-use the data for other works. My recommendation is to provide a summary table of all the data as additional information.*

→ We did not provide individual data as we tried to limit the volume of the study. In revision, we provided the concentrations of all extractable elements measured for bulk and each density fraction for all studied soils as a table (Table A1).

*– The title needs to be changed because evidence that the organic matter of "nanocomposites" has been biotransformed by the microbial compartment is "light". This argument is only based on the value of C:N which shows quite variable values (shown only in the supplementary informations graphs). Even if the trend is probably true, the authors have not investigated enough to state it with certainty.*

→ We agree that we have to be careful on this point. We think the OM in the meso-density fractions is the mixture of both plant- and microbially-derived OM as depicted in Fig. 7b (glad that you liked this!). That was why we did not call it "microbially-derived".

However, we think it is appropriate to call it "microbially-processed" for three reasons: (i) the C:N ratio of the meso-density fractions was consistently lower by 2-23 units relative to the lowest-density fraction which mainly consists of plant detritus. (ii) SEM observation showed decreasing abundance of plant detritus with increasing density, (iii) delta N-15 analysis also showed that the meso-density OM was always more enriched relative to the OM in the low-density fraction by ca 2-6 per mil (unpublished data). The N-15 analysis of the density fractions was done for 14 out of the 23 soil samples. This result is more complex to interpret and will be incorporated into our next work. For these reasons, we think it is

more appropriate to interpret that the majority of the OM present in the meso-density fractions is "microbially-processed".

We revised the abstract section related to this topic as follows (L22-24). "The OM in meso-density fractions showed 2-23 unit lower C:N ratio than the lowest-density fraction of respective soil and thus appeared microbially processed from the original plant material."

*- In the introduction, last paragraph: the scientific questions asked must be made explicit.*

→ We think the questions we asked here are clear enough (one hypothesis and several related ideas).

*- Abstract : - Indicate the regions of origin of the samples.*

→ the information added in Abstract as follows: "We identified density fraction locations of major metal phases and OM using 23 soil samples from 5 climate zones and 5 soil orders (Andisols, Spodosols, Inceptisols, Mollisols, Ultisols) from Asia and North America, including …..".

*- L 24 -27 Sentence too long*

→ We agree that it is a long sentence. We thought hard but we cannot think of a good way to shorten it. We revised slightly to improve the readability.

*- L27 : remove stable : there's no argument that the OM is "stable."*

→ "OM" is now removed.

*Methods - The soils choices could be better justified.*

→ We now explained the rational for the soil samples we selected and expanded the soil sample information including mineralogy and parent material as much as possible (L134-161 in revised ms).

*- Give more details on the fractionation protocol.*

→ Now the protocol is fully described (L 171-182).

*- Explain why the number of fractions is different from one soil to another?*

→ Now it is also explained (L 169-171).

*- SPT is very acidic (pH down to 2); what could be the effect of such a pH on the "nanocomposites" ?*

→ We added the following sentence at the end of density fractionation section under Method.

"We also assume little impact of sodium polytungstate on the extractability of Fe and Al phases or the nature of soil microaggregates as the SPT solution after the density fractionation typically had the pH value similar to bulk soil pH." (L187-188)

*Chemical analyses - Instead of adding weight bas concentration (Al+1/2 Fe) to approximately normalize the atomic mass difference between Al and Fe, working with atomic concentration would be more rigorous!*

→ We debated on this point. We agree that the expression in atomic mass is more rigorous and facilitate the comparison with other materials (e.g., experimental mixture, sediments). But the major interest is the comparison between these metals and C. And the C concentration in soils and soil physical fractions is almost always expressed on weight basis. So we would like to keep our unit as is. But we now also provide these values on atomic mass in appropriate sections (figure legend in Fig. 1 and 3, the main text where we explain OC:metal ratios) to allow such comparison.

So, in the revised manuscript, we added a following sentence in Method section (L214). "This allows us to compare the metal values with C on the weight basis." We also reported some values including the stoichiometric relationships among the target elements (e.g., Al:Si ratio) on molar basis.

*Peak density determination - To my mind, "peak density" is a term which is not really appropriate*

→ We cannot think of any better ways to explain this. Sorry… We edited the main text to enhance clarity on this expression.

*Result - I would suggest to first present the data (see previous remark) before presenting the recovery*

→ We think it is better to show the recovery data first. If the recovery is now good, then the quality of the rest of data becomes questionable. So we prefer to put this information upfront as previous studies did  (e.g., Swanston et al., 2005, Geoderma) https://www.sciencedirect.com/science/article/pii/S0016706104003258

*- L230 Fig A1 : left panel - L290 : I may add an additional sentence to be sure that the reader understand properly the difference between fig 2 concentration and fig 5 distribution*

➔ Thank you for the advice. We added some words to improve the clarity of this first sentence.

*Discussion - L326 what are the "non-centrifugeable colloidal Fe/ Al oxide phases?*

➔ This refers to the colloids that were too small to spin down by the centrifugation used. This corresponds to the limitation of PP extraction discussed 11 lines above this sentence in the same paragraph.

*- L328-331 : no data on Si were provided in the result part. (I think you may add a short section also on Si in the results)*

➔ Thank you for pointing out. We now reported Si results in Result section by reporting (1) Al:Si molar ratios of each density fraction for each soil, and (2) the proportion of total extractable Si in pyrophosphate- and oxalate-extractable phases (Fig. A4). In fact, we expanded the result section (L 299-304) by showing the proportion of total extractable Fe, Al, and Si (M) present as $M_{PP}$ and $M_{OX}$ as Fig. A4.

*- L345 : Is a graph Al:Fe as a function of pH interesting? I would have enjoyed to see it !*

➔ I agree that it would be interesting! But we don't have pH of each density fraction. So we cannot do this.

*- "nanocomposite" could also have been called "nanoCLICS" as proposed by Tamrat et al. 2019.*

➔ We added this acronym where we cited this work in Discussion section.

*- Fig 7a : position of the peaks is not consistent with fig 5*

➔ The Y axis in Figure 7a is the concentration of metal per fraction where as Fig. 5 showed the metal distribution. So Fig. 7a matches with Fig. 2.

We realized that it is easy for readers to get confused with two expressions (concentration and distribution). So we plan to add some sentences to clearly distinguish the two expressions in Method as well as Results/Discussion sections.

*- I do enjoy Fig 7 b !*

→ Very glad that you liked it!

**Anonymous Referee #2**

Thank you for the thorough review and positive comments. Below we show how we plan to address the points raised.

Rota Wagai, on behalf of the authors

*Abstract: I think that the term "density fraction locations" (Line 12) and the sentence in lines 17-18 are hard to understand before the article is read. Please consider alternative phrases.*

→ We agree that "density fraction locations" is not a familiar term but we think it is reasonable because it is logically the same as "size fraction locations".

→ We modified the sentence in line 17-18 to make it easier to understand for readers. The new sentence proposed is as follows.

"The concentrations of Fe and Al (per fraction) extracted by each of the three reagents able Fe and Al concentrations were (per fraction) generally higher in meso-density fractions (1.8-2.4 g cm-3) than in the lower- or higher-density fractions, showing unique unimodal pattern distribution along particle density gradient for each soil and each extractable metal phase. "

*Line 21: Please include that the assumption of having microbial processed OM is based on C/N ratios*

→ Yes, we added the information on C:N ratio.

*Line 27: why is the OM supposed to be stable?*

→ We agree that we don't have enough evidence to state this. We removed the word "stable".

*Manuscript: Line 141: Why were the fractions dried using different methods?*

→ We explained the reason and the assumption behind as follows in Method section.

"The lowest-density fractions (<1.6 g cm-3) were oven-dried at 80 oC instead of freeze-drying for a logistical reason. Due to the concentration of the extractable metals in this fraction, we assumed little effect of the difference in the drying method on our result interpretation.

*Line 291; Figure 6: I wonder why all SEM images look so similar. Everything seems to be aggregated. Is that also the case for the original soils? Or is it possible that the polytungstate treatment promotes aggregation?*

*Figure caption: where do you see clay coatings? Can you add arrows?*

➔ We were also somewhat surprised by the well-aggregated appearance of all these fractions. But the aggregate nature observed by SEM is fairly consistent with the results of other experiments (unpublished) that show that most materials in the meso-density fractions are dispersible only after much stronger energy (by sonication). That was done right after the density fractionation (i.e. the samples remained wet). So we think the aggregated materials we observed are not likely to be artifacts. Clay coatings are the plate-like features you see on most of these aggregates. We plan to upload higher-resolution images for the final version.

➔ We also plan to expand Fig. 6 by including another density fraction (1.8-2.0 g/cc) and adding another soil series (a spodic horizon sample). We are in the process of preparing this. So new Fig. 6 is not reflected in the revised pdf file. I hope it is OK as the additional SEM images will not change the interpretation or the current discussion.

*Line 428: delete "extractable"?*

➔ Thank you for pointing out. Now it was deleted.

*Line 506: correct "understanding"*

➔ corrected.

[revised manuscript text omitted]

---

## Author Response (AR3)

Revision note:

Dear Cornelia,

Thank you for the valuable suggestions. Below we explain how we responded to the three suggestions.

1) We identified density fraction locations of major metal phases and OM using.... could be: we identified the allocation of major metal phases and OM to density fractions....

➔ We changed accordingly.

2) These results led to a hypothesis which involves two distinct levels of organo-metal interaction – the formation of OM-rich, mixed metal phases having relatively fixed OM:metal stoichiometry and subsequent development of meso-density microaggregates via "gluing" action of these organo-metallic phases by entraining other organic and mineral particles such as phyllosilicate clays. - could be replaced by We hypothesised that there are two distict levels of organo-metal interations (1) OM-rich mixed metal phases with fixed OM:metal stoichiometry followed by (2) development by development of meso-density microaggregates via "gluing action of the OM-metallic phases with orther organic and mineral particles such as phyllosilicate clays.

➔ We largely followed your suggestion. In revised version, we wrote as follows (changes in red):

These results led to a hypothesis which involves two distinct levels of organo-metal interaction: (1) the formation of OM-rich, mixed metal phases with relatively fixed OM:metal stoichiometry followed by (2) the development of meso-density microaggregates via "gluing"action of these organo-metallic phases by entraining other organic and mineral particles such as phyllosilicate clays.

3) I suggest that you explain in material and methods why not the same number of density fractions was separated from all soils

In section 2.2 (line 175-177), we now explained the reason as follows (changes in red):

Most soil samples were separated into 6-7 fractions (n=18) while the other 5 samples (A-6, A-7, A-8, A-9, C-1) examined at a later stage were fractionated into only 4 fractions (Table 1, also see Table A1) because we learned that the main allocation pattern can be captured by four density fractions.